# Single-photon advantage in quantum cryptography beyond QKD

Daniel A. Vajner [1,6], Koray Kaymazlar [1,6], Fenja Drauschke [2,6], Lucas Rickert[1], Martin von Helversen[1], Hanqing Liu [3,4], Shulun Li[3,4], Haiqiao Ni[3,4], Zhichuan Niu[3,4], Anna Pappa[2] & Tobias Heindel [1,5] ✉

Quantum key distribution (QKD) can be used to establish a secret key between trusted parties. Many practical use-cases in communication networks, however, involve parties who do not trust each other. A fundamental cryptographic building block for such distrustful scenarios is quantum coin flipping, which has been investigated only in few experimental studies to date, all of which used probabilistic quantum light sources imposing fundamental limitations. Here, we experimentally implement a quantum strong coin flipping protocol using single-photon states and demonstrate a quantum advantage compared to both classical realizations and implementations using faint laser pulses. We achieve this by employing a state-of-the-art deterministic quantum dot light source in combination with fast, random polarization-state encoding enabling sufficiently low quantum bit error ratio. By demonstrating a single-photon quantum advantage in a cryptographic primitive beyond QKD, our work represents a major advance towards the implementation of complex cryptographic tasks in a future quantum internet.

The functionality of today's communication networks relies on diverse, sensitive and complex tasks which, nevertheless, can be built from a set of basic cryptographic building blocks, or primitives. While the security of classical cryptographic implementations relies on assumptions regarding e.g., the computational complexity, it is known that laws of quantum physics can be exploited to enhance the security in communication[1–3]. To date, quantum key distribution (QKD) is by far the most studied quantum cryptographic primitive, enabling unconditional security in the communication between authenticated, trusted parties. Despite its importance and success, QKD is fundamentally limited in its usefulness, as it cannot be used for more complex cryptographic tasks. Many practical use-cases, including randomized leader election, commitment schemes, multi-party computation, or online casinos, involve two or more parties who do not know or trust each other[4–6].

An important example of a cryptographic primitive between two distrustful parties is coin flipping (CF), where two parties toss a coin to choose between two alternatives in the least biased way. Due to the importance of this primitive, M. Blum introduced the idea of 'coin flipping by telephone' already in 1983, in which two spatially separated parties do not necessarily trust each other but still wish to ensure that the outcome of the coin flip is secure[7]. As classical coin-flipping protocols rely on the computational complexity of one-way functions, they can always be broken with sufficient computational power under standard non-relativistic assumptions[8].

This is not the case in the quantum version of coin flipping protocols[9], although a finite bias remains[10,11]. Interestingly, the first quantum coin flipping protocol has been proposed in the very same seminal work by C.H. Bennett and G. Brassard in 1984[2]. While this and many other protocols seemed impractical, more recent proposals

[1]Institute of Physics and Astronomy, Technical University of Berlin, Berlin, Germany. [2]Electrical Engineering and Computer Science Department, Technical University of Berlin, Berlin, Germany. [3]State Key Laboratory of Optoelectronic Materials and Devices, Institute of Semiconductors, Chinese Academy of Sciences, Beijing, China. [4]Center of Materials Science and Optoelectronics Engineering, University of Chinese Academy of Sciences, Beijing, China. [5]Department for Quantum Technology, University of Münster, Münster, Germany. [6]These authors contributed equally: Daniel A. Vajner, Koray Kaymazlar, Fenja Drauschke. ✉e-mail: tobias.heindel@uni-muenster.de

accounted for device imperfections and channel loss, unavoidable in practical scenarios[12,13]. Experimental implementations of quantum coin flipping reported to date used attenuated lasers, i.e., weak coherent pulses (WCPs)[14,15], sources based on spontaneous parametric down conversion (SPDC) exploiting entanglement[16,17], or heralded single photon states[18]. However, as demonstrated in our work, deterministic quantum light sources providing single photons on-demand, can provide an advantage for implementations of cryptographic primitives beyond QKD.

In this work, we experimentally implement a quantum strong coin flipping protocol using an on-demand sub-Poissonian light source and demonstrate its advantage over both classical and WCP-based implementations. To this end, we employ a state-of-the-art single-photon source based on a semiconductor quantum dot deterministically integrated into a high-Purcell micro-cavity in combination with fast dynamic polarization-state encoding with a sufficiently low quantum bit error ratio (QBER), required for the successful execution of this type of protocol. Based on a thorough theoretical analysis, we optimize the protocol parameters and experimentally achieve quantum coin flipping rates on the order of $\approx 1$ kbit/s in a back-to-back configuration. We observe cheating probabilities lower than what is possible in the equivalent classical coin flipping protocol, both in simulations and in the experiment. Moreover, we conducted QSCF experiments under variable attenuation inside the quantum channel and studied its impact on the quantum advantage. Our work represents an important step forward in exploiting quantum advantages in realistic settings for applications in the future quantum internet.

## Results

### The quantum coin flipping protocol

In the context of quantum coin flipping, similar to classical coin flipping, two protocol families, known as quantum strong coin flipping (QSCF) and quantum weak coin flipping (QWCF), are distinguished according to the goals of the two parties. While in QWCF both parties favor a certain outcome (e.g., Alice wants 0 and Bob wants 1, and they are aware of the other party's preference), both parties want a random,

unbiased result in QSCF. Hence, cheating in QWCF must be considered in one direction only, while both directions are important in QSCF, resulting in more protocol constraints for the latter type.

The important figures of merit for a coin flipping protocol are the honest winning probabilities $P_h^A$ and $P_h^B$ of Alice and Bob, their dishonest cheating probabilities $P^A$ and $P^B$, as well as the honest abort probability $P_{AB}$, which describes protocol aborts due to photon loss or errors without cheating. In this context, a protocol is called balanced if Alice and Bob have the same honest winning probabilities ($P_h^A = P_h^B$), and fair when they have equal dishonest cheating probabilities ($P^A = P^B$). Note, that the terms fair and balanced are also used reversely in some work[6]. Here, however, we follow the terminology used in ref. 13. The degree of security of a fair and balanced coin flipping protocol is then measured by the bias $\epsilon = \max[P^A, P^B] - \frac{1}{2}$, that quantifies how much the dishonest cheating probability differs from the case of random guessing with probability 1/2. The minimum theoretically achievable bias for a QSCF protocol was shown by Kitaev to be $\epsilon \le \frac{1}{2}(\sqrt{2}-1) \approx 0.21$[19], which corresponds to a minimum cheating probability of 71%. On the other hand, QWCF protocols with arbitrarily small bias exist[20], which are, however, often based on assumptions difficult to achieve experimentally. More recently, a newly proposed QWCF protocol[21] reaching Kitaev's bias limit was also successfully implemented experimentally[18]. Note that a successful implementation of QSCF always trivially implies a QWCF version of the same bias, as the two parties can simply place bets on the outcome of the random bit.

The protocol chosen for the implementation in this work is one of the most established ones in the family of QSCF protocols and has been proposed in ref. 12 and extended in ref. 13. Previously, this protocol was used to show an experimental quantum advantage using attenuated laser pulses[15]. In the following, we first introduce the QSCF protocol and discuss on the basis of simulations how single photons can be exploited to enhance its quantum advantage, before we turn to the experimental implementation.

The main steps of the QSCF protocol from ref. 13 implemented in this work are the following[15] (cf. Fig. 1):

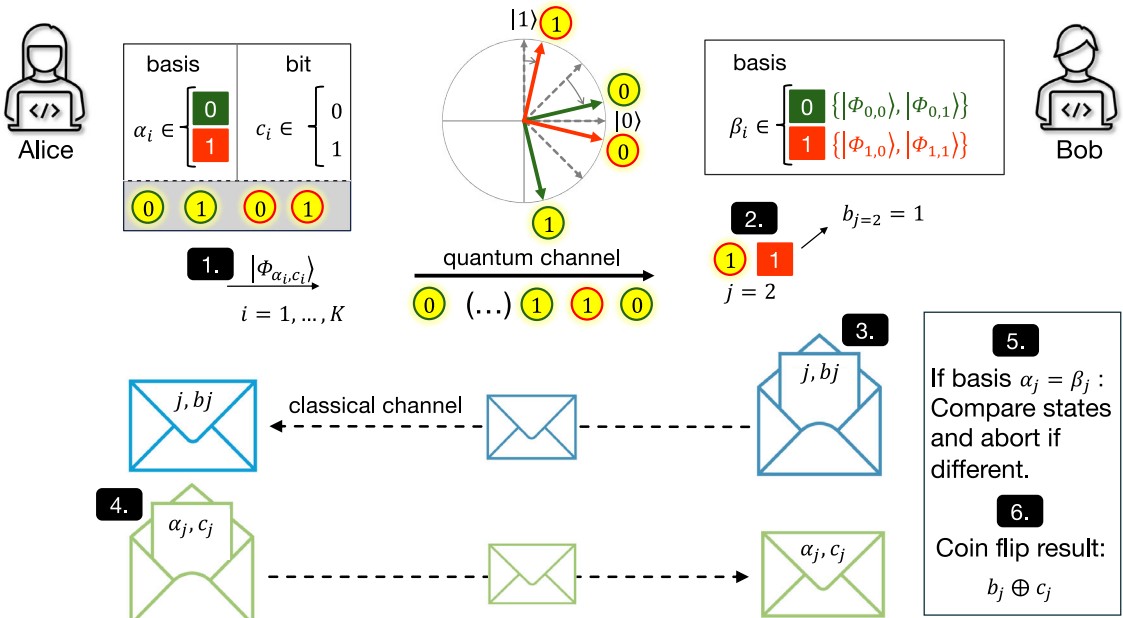

**Fig. 1 | Schematic of the QSCF protocol implemented in this work.** 1. Alice randomly prepares pulse number $i$ of the $K$-long sequence in one out of four pre-optimized coin flipping states $\left|\phi_{c_i, c_i}\right\rangle$ and sends them to Bob. The states are rotated relative to the standard BB84 states (dashed gray lines). 2. Bob projects the received pulses randomly into one of the same four states and calls the first detected event $j$. 3. Bob now sends a random number $b_j$ and the pulse number $j$ to Alice. 4. Alice returns her initial bit $c_j$ and basis $\alpha_j$ for that pulse. 5. If Bob measures a different state for the same basis, he aborts. 6. If the protocol is not aborted the coin flip result is $c_j \oplus b_j$.

1. In each step $i$ of a $K$-step long protocol round, Alice prepares and sends pulse $i \in (1, K)$ to Bob, in a randomly chosen preparation basis $\alpha_i \in \{0, 1\}$ and encoding bit value $c_i \in \{0, 1\}$. The four possible states $|\phi_{\alpha_i, c_i}\rangle$ read:

$$\left|\phi_{\alpha_i, 0}\right\rangle = \sqrt{a}|0\rangle + (-1)^{\alpha_i}\sqrt{1-a}|1\rangle \text{ and}$$
$$\left|\phi_{\alpha_i, 1}\right\rangle = \sqrt{1-a}|0\rangle - (-1)^{\alpha_i}\sqrt{a}|1\rangle, \tag{1}$$

   with the real number $a \in (0.5, 1)$, used to fine-tune the initial states to optimize the protocol and ensure equal cheating probabilities of Alice and Bob.

2. Bob picks a random measurement basis $\beta_i$ for each pulse. If Bob has not detected any of the $K$ pulses, the protocol aborts, otherwise, the first detection event is $j$.

3. Bob picks a random number $b_j$ and sends it to Alice via a classical channel together with the index $j$.

4. Alice reveals her corresponding basis and bit value $(\alpha_j, c_j)$ to Bob.

5. Whenever the same basis was used $(\alpha_j = \beta_j)$, Bob confirms whether he measured the same state $\left|\phi_{\alpha_j, c_j}\right\rangle$ or aborts if the states disagree.

6. If Bob has not aborted, the outcome of the coin flip is $c_j \oplus b_j$

The probability of aborting when both parties do not attempt to cheat $P_{AB}$ depends on different protocol parameters such as the number $K$ of pulses sent per coin flip, the transmission $\eta_{Bob}$ inside Bob's receiver setup, the photon detection efficiency $\eta_{Det}$ of Bob's detector, the detection error $e$, the mean number of photons per pulse $\mu$ and the photon statistics of the employed light source.

It is important to note that the photon statistics of the employed light source crucially affect the protocol performance, as discussed in the following. Phase-randomized weak coherent pulses follow a Poisson distribution, i.e., the probability $p_n$ that a pulse contains $n$ photons depends on the mean photon number $\mu$ according to $p_n = \frac{e^{-\mu}\mu^n}{n!}$. In contrast, the photon statistics of a deterministic single-photon source, as used in our implementation, is limited by experimental imperfections only. In this case, the amount of multi-photon contribution can be upper-bounded via the anti-bunching value $g^{(2)}(0)$ as ref. [22]

$$p_1 \approx \mu, p_2 \leq \frac{1}{2}\mu^2 g^{(2)}(0), p_0 \approx 1 - p_1 - p_2. \tag{2}$$

## Protocol performance simulation

Next, we simulate the performance of the QSCF protocol to evaluate the optimal parameters for our implementation. Accounting for the experimental conditions in our protocol implementation, the cheating probabilities can be calculated as a function of the protocol parameter $a$ defining the tilt-angle in the four coin flipping states (cf., Eq. (1)). Optimizing the parameter $a$ one can minimize the cheating probabilities, while maintaining $P^A = P^B$, for each parameter from the set $\{\mu, K, \vec{p}\}$ (see "Methods" and Supplementary Note 3). This optimization is performed for the parameters summarized in Table 1 corresponding to the experimental conditions realized in our implementation further below. Figure 2a shows the obtained optimized cheating probabilities as a function of the number of rounds $K$ for a QSCF protocol implemented with weak coherent pulses (WCP, cf. blue line), a single photon source (SPS, cf. green line) and the equivalent classical protocol (black line). A quantum advantage is achieved when the QSCF cheating probability is smaller than in the classical protocol (blue shaded region). We find that the SPS implementation outperforms the WCP version in the full parameter range, while at high $K$ values only the SPS can achieve a quantum advantage (green shaded region). The dashed green line indicates an ideal SPS without any multi-photon pulses $(g^{(2)}(0) = 0)$. In this case, no 2-photon events can be used to cheat, thus preserving the quantum advantage for large $K$. However, for $K \ll 1/\mu$,

## Table 1 | Parameters used in the QSCF protocol simulations, corresponding to the experimental conditions

| Parameter | Value |
|---|---|
| Detection module transmission $\eta_{Bob}$ | 0.5 |
| Detector efficiency $\eta_{Det}$ | 0.85 |
| QBER $e$ | 0.028 |
| Anti-bunching value $g^{(2)}(0)$ | 0.03 |
| Mean photon number $\mu$ | 0.0013 |
| Protocol rounds $K$ | $5 \times 10^4$ |
| State parameter $a$ | 0.9 |
| System clock rate $R_0$ | 80 MHz |
| Dark-count probability $P_{dc}$ | $4 \times 10^{-7}$ |

the probability of multi-photon events becomes negligible even for a non-ideal single-photon source, which is why the maximum quantum advantage is the same, and even an ideal single-photon source would not further reduce the cheating probability for this protocol and experimental parameters.

An extended comparison between SPS and WCP implementation confirms that both provide a quantum advantage for realistic parameters, while the advantage is higher for the SPS. Figure 2b, c illustrates the difference between classical cheating probability and QSCF cheating probability. Here, regions of positive values identify a quantum advantage. The simulations reveal that in implementations using realistic single-photon sources, the reduced multi-photon contributions enable a quantum advantage in a larger parameter range compared to WCPs. Note, that the advantage of our SPS compared to WCPs can exceed 8% for larger $K$ (see Supplementary Note 4). This, however, occurs in a regime with a reduced advantage compared to the classical case. In our work, we therefore chose parameters that clearly yield a quantum advantage while simultaneously enabling a single-photon advantage.

In this context, two aspects of QSCF are noteworthy compared to QKD: Firstly, the concept of decoy-states for mitigating multi-photon effects, as known from laser-based QKD[23], cannot be employed in the distrustful setting of coin flipping. Hence, all multi-photon events increase the cheating possibilities for Bob. Secondly, achieving a quantum advantage in the QSCF protocol is more sensitive to bit errors. For the chosen protocol, QBERs below $\approx 4\%$ are required to achieve a quantum advantage (see Methods), while BB84-QKD can tolerate up to 11% QBER and still asymptotically generate a secure key[24]. The reason is that in QKD, errors can be mitigated during post-processing, while they reduce the equivalent classical cheating probability in QSCF due to an increased number of protocol aborts. Note also that the employed cheating probability bound for Bob in the QSCF protocol is not tight, and lower bounds might be found in the future, which would increase the effective quantum advantage.

One could argue that the same cheating probabilities and almost the same quantum advantage of the single-photon source could, in principle, be reproduced using a laser with a smaller mean photon number $\mu$, as the likelihood of multi-photon events is proportional to $\mu^2$. However, a smaller $\mu$ would require higher $K$ values, leading to a lower rate of successful coin flips, as the maximum rate of secure coin flips is $R_{CF} = R_0/K \sim R_0 \cdot \mu$, where $R_0$ is the system clock rate. Therefore, employing sub-Poissonian light sources in QSCF provides not only a reduced bias but also a better performance for the same bias.

## Experimental setup overview

To implement the QSCF protocol simulated above, we realized the experimental setup shown in Fig. 3a. Alice uses a deterministic single-photon source to generate flying qubits at a clock rate of $R_0 = 80$ MHz

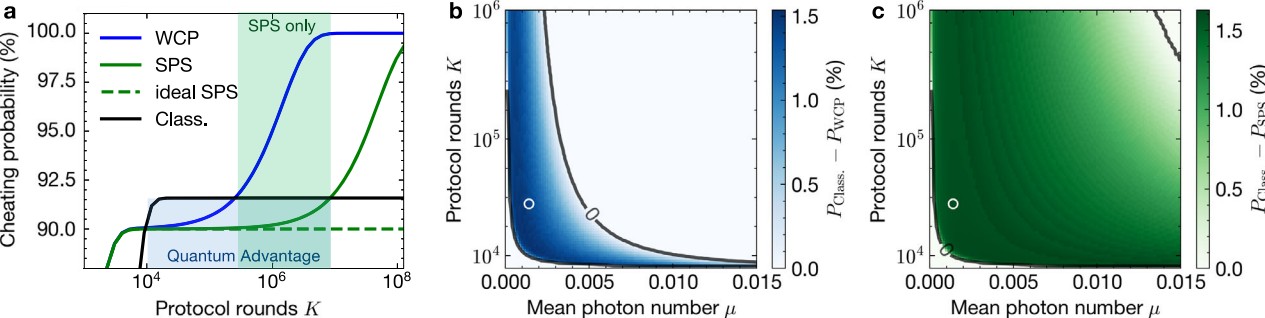

**Fig. 2 | Cheating probabilities for the QSCF protocol. a** Plotted as a function of the number of rounds $K$ per coin flip for implementations using a single-photon source (SPS, green), weak coherent pulses (WCP, blue), and an equivalent classical implementation (Class., black) assuming $\mu = 0.0013$. An ideal single-photon source without multi-photon pulses is shown as a reference (dashed green line). **b, c** Reduction of cheating probability compared with classical protocol as a function of $K$ and $\mu$ achievable in the QSCF protocol implemented with an attenuated laser and a sub-Poissonian light source ($g^{(2)}(0) = 0.03$), respectively. White circles mark the operating point of our experiment.

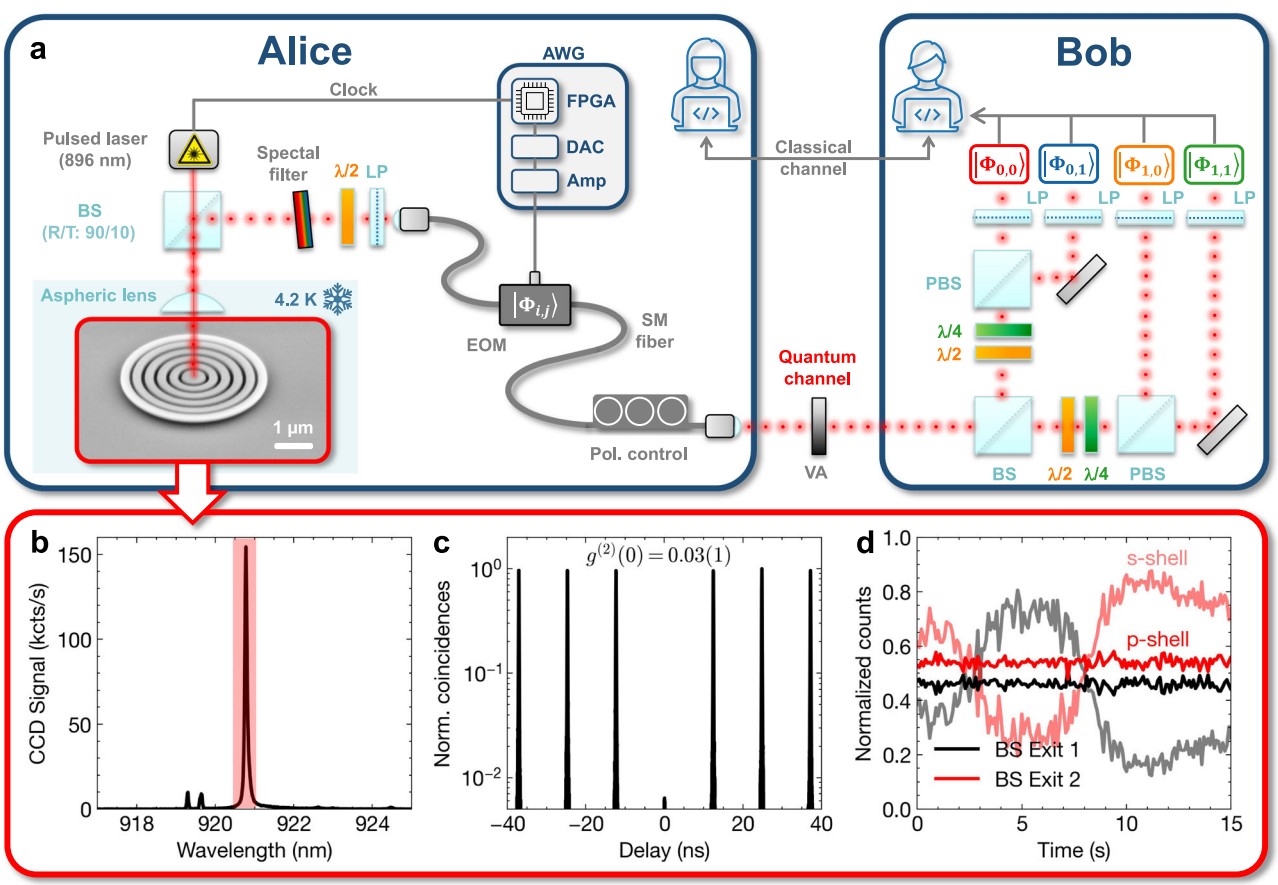

**Fig. 3 | Experimental quantum coin flipping using polarization-encoded single-photon states. a** Experimental setup for QSCF: Alice generates single photons using a quantum dot microcavity and randomly switches their polarization state between the four protocol states using a fiber-coupled electro-optical modulator (EOM) controlled by a self-built arbitrary waveform generator (AWG). The polarization-encoded qubits propagate through a quantum channel and are detected in a 4-state polarization analyzer with a passive bases choice on Bob's side. (BS: beam splitter, LP: linear polarizer, PBS: polarizing BS, DAC: digital-to-analog converter, FPGA: field-programmable-gate-array, Amp: Amplifier, SM: single-mode). **b** Emission spectrum of the single-photon source and (**c**) Hanbury-Brown and Twiss experiment confirming the single-photon nature of the spectrally filtered emission from (**b**) (red shaded area). **d** Measurement of the photon number coherence using phase-resolved two-photon interference experiments under p-shell (dark lines) and strict resonant (light lines) excitation. As QSCF requires vanishing photon number coherence, quasi-resonant excitation was chosen for the protocol implementation in this work.

and dynamically switches randomly between the four different polarization-encoded QSCF states in a fiber-coupled electro-optical modulator. Next, the polarization qubits propagate through a short free-space optical channel, including a variable attenuator for emulating channel loss. The single-photon pulses are detected by Bob

using a 4-state polarization analyzer with passive basis choice in combination with superconducting nanowire detectors and time-tagging electronics (see "Methods"). Classical post-processing is performed via a classical data link. The execution of the QSCF protocol relies on two building blocks, namely the single-photon generation

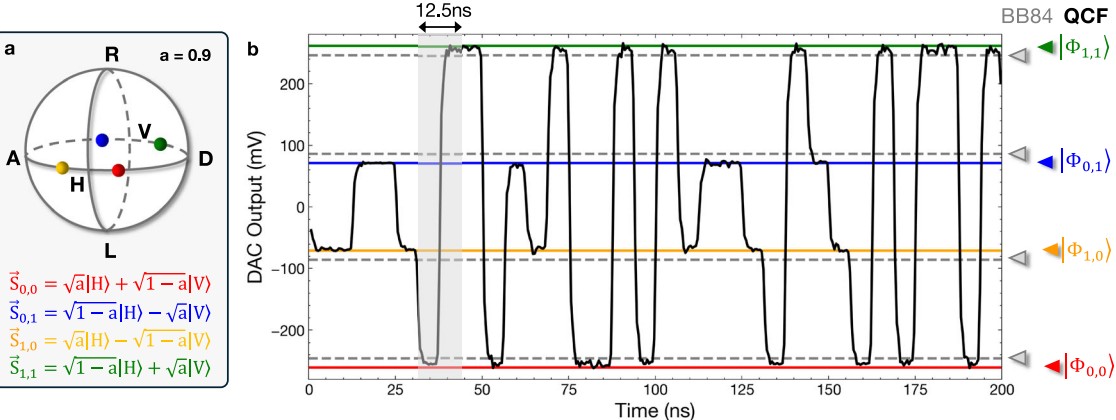

**Fig. 4 | Creating the polarization states. a** The four states used in the QSCF protocol are defined by the parameter $a = 0.9$ and marked on the Poincaré sphere. **b** Exemplary sequence showing random voltage-level switching as used in our protocol implementation to modulate the EOM for dynamic polarization qubit encoding. Full horizontal lines indicate the four target states of the QSCF protocol. Dashed lines indicate the states typically used in BB84 QKD $\{H, D, V, A\}$ serving as a reference. Note that we employed an advanced coding scheme requiring an effectively doubled clock rate of 160 MHz inside the control electronics, preventing voltage level drifts as described in the main text. The gray shaded window indicates the 12.5 ns wide period defined by the 80 MHz laser clock-rate, where the polarization encoding of the photon is performed during the first 6.25 ns of the window.

and the dynamic qubit encoding on Alice's side, as discussed in the following.

### Flying qubit generation

The single-photon source comprises a single pre-selected semiconductor quantum dot emitting at a wavelength of 921 nm, deterministically integrated into a micro-cavity based on a hybrid circular Bragg grating (cf. inset in Fig. 3a) to enhance the photon extraction efficiency and reduce the radiative lifetime to 50 ps via high Purcell enhancement. For details on the design, deterministic fabrication and in-depth quantum-optical characterization of this type of single-photon source, we refer to ref. 25 and ref. 26, respectively. Choosing quasi-resonant (p-shell at 896 nm) optical excitation of the quantum emitter operated in a cryogenic environment (4 K) results in the emission spectrum in Fig. 3b. Here, the predominant emission of a charged state used for our experiments in the following is identified. Performing a Hanbury-Brown and Twiss measurement after spectral filtering via a monochromator and coupling to a single-mode fiber confirms the single-photon nature of the emission with an uncorrected and integrated anti-bunching value of $g^{(2)}(0) = 0.03(1)$ (cf. Fig. 3c). Furthermore, it is essential for the protocol implementation that the emitted photon states possess no coherence in the photon number basis, a necessary assumption in the treated cheating strategies, as recently discussed by Bozzio et al.[27]. This was confirmed for the excitation conditions used in the coin flipping implementation, by interfering sequentially emitted single photons in a Mach-Zehnder-interferometer and varying their relative phase while monitoring the countrate after the interference. The absence of oscillations observed in this experiment, as seen in Fig. 3d confirms that the photon number coherence in the used light state is negligible, thus, the analytical formalism based on ref. 15 can be applied without the need for active phase-randomization. As a cross-check, we repeated the experiment with the same quantum emitter under strict resonant excitation ($\approx 0.2\pi$ pulse area), revealing oscillations due to a finite photon number coherence, as expected for this excitation scheme.

### Dynamic polarization state encoding

Dynamic state preparation on Alice's side is performed using a fiber-coupled EOM controlled by a self-built arbitrary waveform generator based on field-programmable gate-array (FPGA) electronics, digital-to-analog converters (DAC) and an amplifier (Amp). The polarization

encoded qubit states leaving the EOM are then rotated into the final protocol states via fiber polarization paddles just before entering the quantum channel. Figure 4a illustrates the desired qubit states and the corresponding polarization Stokes vector $\vec{s}$ on the Poincaré sphere. Note that the voltage levels corresponding to the desired polarization states need to be carefully adjusted, due to the relatively small difference to the standard BB84 states. To achieve the lowest QBER in our protocol implementation, we employed an advanced coding scheme, which required a doubling of the effective clock rate inside the AWG to 160 MHz. While this made the precise timing control more demanding, it effectively suppressed voltage level drifts in random state sequences, enabling low QBER levels with improved temporal stability (see "Methods"). Figure 4b shows an exemplary random voltage level sequence used as input for the EOM in our protocol implementation, where alternating voltage modulation was applied. This measurement confirms the correct adjustment of the voltage levels to the four target states for our implementation (full lines). Please note the small but crucial difference between the QSCF states used in this work, corresponding to $a = 0.9$, and the BB84 states typically used in QKD (dashed gray lines). Using the encoding system presented above, we experimentally achieve an overall QBER for dynamic random state switching of single-photon pulses of 2.8%. In order to achieve a quantum advantage, a QBER below $\approx 4\%$ is required for the protocol chosen here (see Methods), as a higher QBER would lead to a reduced classical cheating probability due to many honest aborts.

### Experimental quantum strong coin flipping

To implement the QSCF protocol, Alice randomly encodes the four protocol states for the fixed optimized value of $a = 0.9$ and a fixed number of pulses $K = 50,000$. After the $K$ voltage values, the sequence is repeated to evaluate the protocol performance with sufficient statistics. To implement the random bit sequence, we used a pre-stored set of quantum random numbers generated by measuring vacuum fluctuations[28,29], which were provided by the Australian National University[30]. After propagation through the quantum channel, first in the back-to-back case without additional loss, Bob projects the arriving photons into the four protocol states by detecting them in his four detection channels, yielding the arrival statistics shown for a short time period in Fig. 5a. Synchronizing the qubit-state preparation and detection using a trigger signal at the start of each protocol run, we are able to average over many realizations of the $K$-long random sequence.

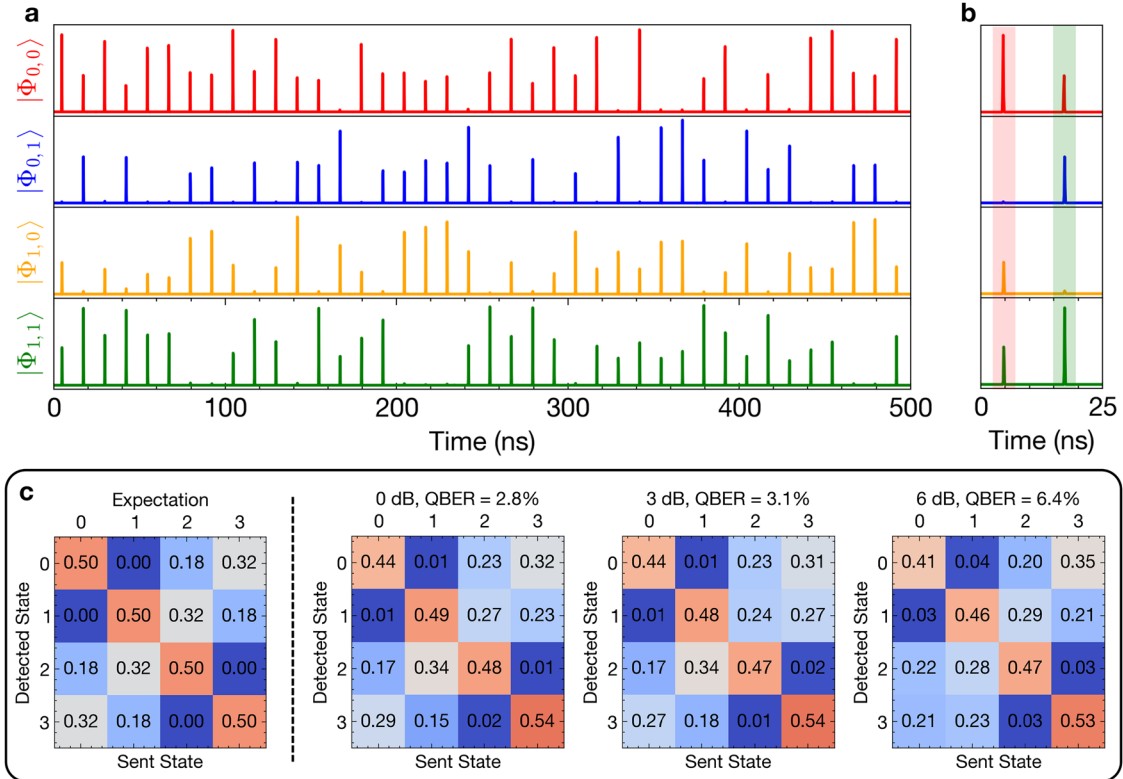

**Fig. 5 | Analyzing single-photon states. a** Time traces of single-photon pulses under dynamic random switching between the four QSCF states as detected on Bob's side after projection into the four different target channels (top to bottom). **b** Zoom-in from (**a**) with shaded regions indicating the prepared state by the color coding. **c** Integrating over all coin flip realizations and comparing prepared and detected states yields input-output matrices from which the QBER can be computed. Introducing additional loss in the quantum channel, the QBER increases, and the received statistics start to differ from the expected case for $a = 0.9$ (left panel).

The relative temporal offset between Alice' and Bob's time-bins is determined by correlating a set of the sequence for different relative shifts and channel permutations. This enables us to compare the detected state to the actually sent state for each time bin (see Fig. 5b), which yields a high extinction in the preparation basis and the expected projections in the other bases. Figure 5c presents the resulting input-output matrices, as theoretically expected from the parameter $a$ (left panel) and for the experimental realizations with losses in the quantum channel of 0 dB (back-to-back), 3 dB, and 6 dB, resulting in a QBER of 2.8%, 3.1%, and 6.4% (right panels). The QBER obtained under dynamic random state preparation was calculated from the ratio of wrong projections divided by all projections for a given state and averaged over all four QSCF states. We observe a good agreement between the experimental performance and the theoretical expectations.

Performing the protocol steps outlined above, coin flips are performed for each realization of the $K$-step long random sequence for the back-to-back case, yielding the performance summarized in Table 2. We perform 52,978 successful coin flips within 34 s, yielding a rate of about 1500 secure single-photon coin flips per second. Honest aborts due to sequences in which no photon is detected are almost negligible for the chosen length of $K$. Honest aborts due to deviations between Alice and Bob's outcomes, however, appear in $P_{AB} = 1.4\%$ of the cases, in good agreement with the theoretical prediction that the honest abort probability converges to $e/2$, as such an error is detected in 1/2 of the cases in the protocol. Using the $a$ parameter and the photon statistics, we can calculate the maximum cheating probabilities of Alice and Bob to be $P^A = P^B = 90.0\%$, confirming that the protocol is fair, while the coin flip outcome probabilities of $P_0 = 49.9\%$ and $P_1 = 50.1\%$ confirm the assumption of a random basis choice and the correct generation of a random bit. From the measured honest abort probability $P_{AB}$ we can

further estimate the equivalent cheating probability for a classical coin flipping protocol, resulting in 91.6%. The smaller cheating probabilities observed for the quantum version of the coin flipping protocol demonstrate that we indeed experimentally achieved a quantum advantage. Moreover, the same protocol carried out with a phase-randomized WCP source of the same $\mu$ using the same experimental setup would yield calculated cheating probabilities of 90.3%, higher than in the single-photon case. Hence, we demonstrate not only a quantum advantage, but also a single-photon advantage compared to WCP-based implementations, as predicted in the protocol simulations.

Next, we investigate the performance of our QSCF implementation as a function of additional transmission losses of 3 dB and 6 dB in the free-space quantum channel, corresponding to several km of fiber transmission representative for realistic urban quantum networks. To compensate for the decreasing number of photons detected in each $K$-long sequence with increasing transmission loss, we gradually increased $K$ from now $K = 25,000$ in the lossless case up to $K = 100,000$ for 6 dB attenuation. Figure 6a depicts the resulting honest abort probability $P_{AB}$ as a function of the channel loss. We find that despite the increase in $K$, the honest abort probability still increases due to the reduced signal-to-noise ratio, causing an increased detection error. The increase in $P_{AB}$ quantitatively matches the theoretical expectation (solid lines), as calculated from the experimentally measured QBER $e$ and the selected length of the random sequence $K$.

The experimental quantum advantage achieved in our protocol implementation for different channel attenuations is presented in Fig. 6b, as calculated from the protocol parameters and the measured honest abort probabilities (points), as well as from the theoretically predicted honest abort probabilities (solid lines). We observe that a quantum advantage is maintained for 3 dB additional loss in the quantum channel, while no quantum advantage is present at 6 dB loss

**Table 2 | Summary of the performance of our QSCF protocol implementation obtained in the back-to-back case (0 dB loss)**

|  | SPS Predicted | SPS Implemented | WCP Calculated | Classical Protocol |
|---|---|---|---|---|
| Honest abort prob. $P_{AB}$ | 1.4% | 1.4% | 1.4% | – |
| Bob cheating prob. $P^B$ | 90.0% | 90.0% | 90.3% | 91.6% |
| Outcome '0' prob. $P_O$ | 50.0% | 49.9% | – | – |
| Quantum Gain $g$ | 1.6% | 1.6% | 1.3% | – |

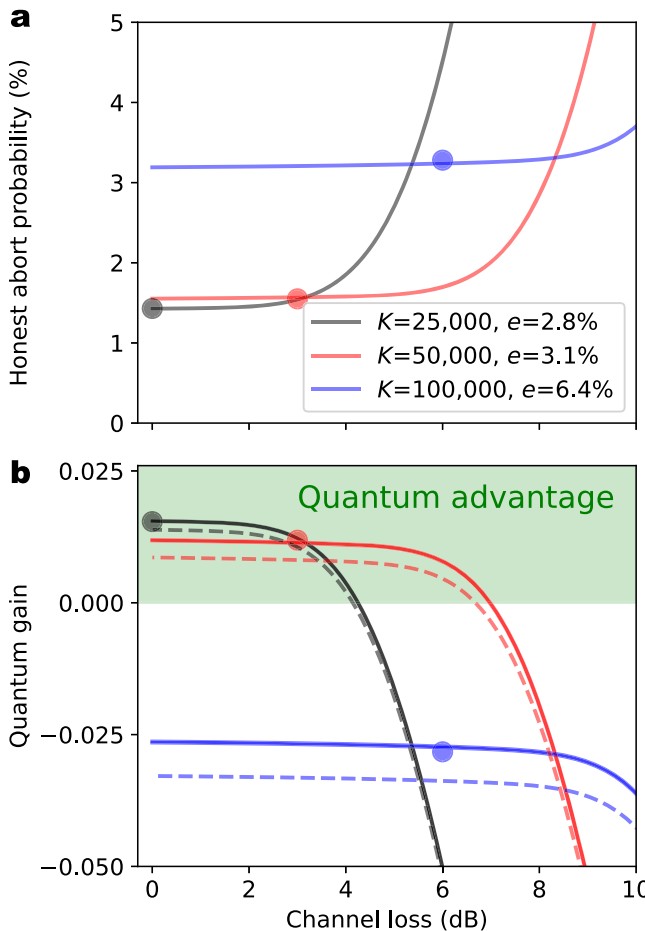

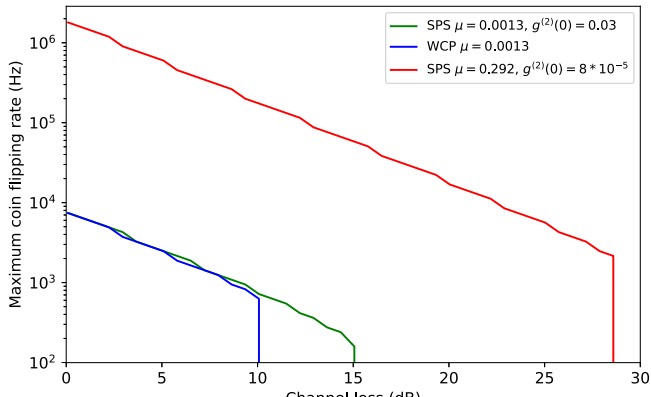

**Fig. 7 | Maximally achievable rate of secure coin flips as a function of channel loss.** Rates are calculated while maintaining a positive quantum advantage and assuming the experimental parameters of the single-photon source used in this work (green line), an equivalent attenuated laser source (blue line), and an realistically improved single-photon source (red line) combining state-of-the-art metrics achieved in refs. 31,32.

**Fig. 6 | Performance of the QSCF implementation. a** Experimentally determined honest abort probability (points) shown as a function of emulated noise increases due to no detections and wrong detections, matching the theoretical expectation (solid lines) for the same $K$ and QBER. **b** Quantum gain (difference between classical and quantum cheating probability) as calculated from the experimentally measured honest abort probabilities (points), errors and protocol parameters, shown together with the theoretical prediction (solid line). The equivalent attenuated laser pulse source (dashed lines) would lead to a smaller quantum advantage (highlighted as a green region). In the high-loss regime the quantum gain vanishes due to the increasing honest abort probability (because of higher errors and less detection events), which makes classical cheating more difficult. The length $K$ of the single-photon sequence transmitted was gradually increased to compensate for the reduced number of detection events.

due to the increased detection errors, as well as a non-optimal choice of $K$ (cf. discussion further below). The experimental results are again in good agreement with the theoretical predictions for the same parameters. The decreasing quantum advantage with increasing loss directly results from the increase in honest abort probability, causing the equivalent classical cheating probability to decrease. In addition, the single-photon advantage is illustrated by plotting the equivalent WCP quantum advantage (dashed lines), indicating that the advantage

becomes more relevant for higher loss, where the need for a higher $K$ increases the influence of multi-photon pulses.

To further explore the effect of loss, Fig. 7 shows the maximum possible rate of secure coin flips $R$, which is calculated from the system clock rate (80 MHz) divided by the smallest possible value of $K$ still yielding a positive quantum advantage. Here, we assumed that the error at zero attenuation is $e_0 = 2.8\%$ and modeled the increase in error with loss (see Methods). By comparing our single-photon source (green line) to the equivalent attenuated laser source (blue line), we confirm that the sub-Poissonian photon statistics lead to performance improvements. Assuming an improved implementation (red line), using a single-photon source that combines the best values for $\mu = 0.29$ after dynamic state preparation[31] and $g^{(2)}(0) = 0.0001$[32] achieved with quantum dots to date, would lead to substantial performance improvements in single-photon QSCF. Here, the higher source efficiency $\mu$ allows shorter protocol sequences $K$, which in turn enable higher coin flipping rates, while a larger $K$ can be tolerated at higher loss when $g^{(2)}(0)$ is smaller.

## Discussion

In this work, we have demonstrated, both theoretically and experimentally, that single-photon sources can be used to achieve a quantum advantage for an essential cryptographic primitive beyond QKD. We implemented a QSCF protocol using a deterministic single-photon source based on a semiconductor quantum dot embedded in a high-Purcell photonic micro-cavity in combination with dynamic random polarization-state encoding with a QBER of 2.8%. The implementation enabled us to experimentally achieve a quantum advantage of up to 1.8% (1.6% percentage points) compared to an equivalent classical realization of the protocol. In addition, we also verified a noticeable single-photon advantage compared to an implementation using faint laser pulses. The experimentally obtained performance agrees well with the theoretical predictions extracted from the simulations. The

observed single-photon advantage is a direct result of the sub-Poissonian photon statistics of the quantum light source used in our experiment. Moreover, we conducted QSCF experiments under variable attenuation inside the quantum channel with different fixed values of $K$, revealing that a quantum advantage can be maintained up to 3 dB of additional loss in our experiment. Calculations of the maximum achievable coin flipping rate show that a quantum advantage of up to 15 dB loss is possible in principle for our experimental parameters using optimized values of $K$. Using single-photon sources operating in the telecom C-band (or lower wavelength emitters using frequency conversion) in future experiments will enable secure quantum coin flipping over distances of several tens of kilometers.

Our protocol implementation paves the way for further advancements in experimental QSCF in future work. Firstly, the QBER can be reduced further using a high extinction ratio polarization-maintaining fiber at the EOM input or alternative concepts for encoding[33]. Moreover, pushing the clock-rate of our implementation from 80 MHz to the GHz-range, as readily possible with our current single-photon source[26], the quantum coin flipping rate can be increased by a factor ×16. This, however, will require a speed-up of the qubit-state encoder, currently limited to an electronic bandwidth of about 300 MHz. Furthermore, transferring our QSCF protocol implementation to telecom wavelength will significantly increase the possible communication distance in optical fibers. Additional performance improvements are possible by using single-photon sources with higher efficiency as well as optimized, loss-dependent choices of $K$. Finally, employing techniques for the direct fiber-pigtailing of deterministic quantum light sources[34] for the integration in compact cryocoolers, will enable the realization of bench-top server-rack compatible quantum coin flipping systems for field experiments as previously demonstrated for QKD[35].

While the quantum advantage demonstrated in this work is still relatively moderate, due to the general and assumption-free protocol used, we anticipate that our work stimulates further research in new directions of secure quantum communication and computation. For instance, implementing alternative strong[36] or weak[37] coin-flipping protocols, the achievable quantum and single-photon advantage can be comparatively studied for different protocols. Furthermore, it would be interesting to explore how single-photon sources can enhance the performance of protocols in restricted adversarial settings. Examples are scenarios in which the communicating parties have imperfect memories[38,39] or restricted measurement capabilities[40,41], potentially allowing one to achieve cheating probabilities much closer to 50%.

## Methods

### Analytical expression for honest abort probability

The honest abort probability $P_{AB}$ can be calculated from the probability $Z$ that a pulse of Alice does not cause a detection event at Bob, the probability $P_{dc}$ that Bob's detector clicks, even though no pulse arrived, and the probability $e$ that a state is detected incorrectly (referred to as QBER above). According to ref. 13 the honest abort probability $P_{AB}$ can thus be expressed as follows:

$$P_{AB} = Z^K (1 - P_{dc})^K + \left[ \sum_{i=1}^{K} (1 - d_B)^{i-1} P_{dc} Z^i \right] \cdot \frac{1}{4}$$
$$+ \left[ 1 - Z^K (1 - P_{dc})^K - \sum_{i=1}^{K} (1 - P_{dc})^{i-1} P_{dc} Z^i \right] \cdot \frac{e}{2} \quad (3)$$
$$\text{with } Z = p_0 + (1 - p_0)(1 - \eta).$$

Here, $\eta = \eta_{Bob} \cdot \eta_{Det}$ is the product of the transmission of Bob's receiver module and the detection efficiency and $p_n$ the probability that a pulse contains $n$ photons. In the case of phase-randomized attenuated laser pulses with a mean photon number $\mu$, $p_n$ follows a Poisson distribution $p_n = \frac{e^{-\mu}\mu^n}{n!}$, while for deterministic single-photon source $p_n$ is only

limited by experimental imperfections and can be approximated by Eq. 2[22].

From the honest abort probability, the classical cheating probability can be calculated as $P_C \leq 1 - \sqrt{P_{AB}/2}$[42], which is inversely proportional to $P_{AB}$. While the honest abort probability is not sensitive to the photon statistics, the main difference between WCPs and single-photon sources lies in the cheating probabilities. Cheating is possible when an error occurred without an honest abort. This case becomes more likely at higher $\mu$, since there are more multi-photon pulses and an abort is less likely. Consistently, sending more photons makes an abort less likely and hence classical cheating possible. See Supplementary Note 1 for a comparative graphical discussion of the honest abort probability $P_{AB}$ and the classical cheating probability $P_C$ as a function of $\mu$ and $K$ (complementing main Fig. 2a).

### Analytical expressions for cheating probabilities

The security analysis follows refs. 13,15, based on ref. 43 for the original protocol[9]. Alice's optimal cheating strategy consists of sending entangled states, waiting for Bob's announcement, and then performing a measurement on her part of the state, which in the worst case results in a cheating probability of ref. 13

$$P^A \leq \left[ 3 + 2\sqrt{a(1-a)} \right]/4. \quad (4)$$

This leads to a maximum cheating probability of 100% for $a = 0.5$ (cf. Supplementary Fig. S3). For Bob, the optimum cheating strategy depends on whether he has detected 0, 1, or 2 photons, assuming that he can differentiate the photon number. These cases are (cf. ref. 13):

- ($A_1$) Bob does not detect a single photon
- ($A_2$) Bob detects at least one 1-photon pulse
- ($A_3$) Bob detects exactly one 2-photon pulse
- ($A_4$) Bob detects exactly one 2-photon and at least one 1-photon pulse

Summing the contributions leads to an upper bound for Bob's total cheating probability of ref. 13

$$P^B \leq \sum_{i=1}^{4} P(A_i) \times P(b'|A_i) + \left[ 1 - \sum_{i=1}^{4} P(A_i) \right] \cdot 1, \quad (5)$$

according to ref. 15 (see Supplementary Table 1). For simplicity, Bob is assumed to be able to cheat with unity probability in all other cases $A_{i>4}$. Note that the cheating probability bound defined for Bob is not tight, and lower bounds are possible in principle. However, while more explicit events could be considered to further tighten Bob's bound, the probability for higher-order multi-photon events is not only very low, but also their cheating probability is very close to the conservatively assumed 100%. In the reasonable parameter regime of $K \approx 1/\mu$ used in our work, which allows for practical coin flipping rates, the probability of all additional cases $A_{i>4}$ can be estimated to be $< 10^{-8}$ for attenuated laser pulses and $< 10^{-12}$ for our QD light source. Hence, we consider the impact of terms $A_{i>4}$ negligible for our study. Note also that if Alice were to send laser pulses without phase-randomization, Bob could cheat better due to the remaining photon number coherence, which would require a refined analysis[27]. See Supplementary Note 2 for an extended and comparative graphical discussion on Bob's cheating probability for WCPs and realistic single-photon sources as a function of $\mu$ and $K$, as well as the classical cheating probability bound.

### Estimation of lower error limit

A quantum advantage is achieved when the dishonest cheating probability of the fair and balanced protocol is smaller than the classical cheating probability, which is determined by the honest abort probability. The smallest honest abort probability is achieved in the regime

of large $K$, where no aborts are due to non-detections, and aborts take place only due to errors which are detected in half of the cases, thus $P_{AB} \geq e/2$. This translates into a classical cheating probability of $P_C \leq 1 - \sqrt{e/4}$. On the other hand, the smallest possible cheating probability in the regime in which photons are detected is $a$. Hence, a quantum advantage requires approximately

$$e \leq 4(1-a)^2 \approx 4\% \tag{6}$$

### Simulations for parameter optimization

To maximize the quantum advantage while ensuring a fair protocol in our implementation, we performed simulations to optimize the parameters used in the experiments. To this end, we minimized the cheating probabilities under the constraints of achieving a quantum advantage ($P_{quantum} < P_{classical}$) and $P^A = !P^B$. The optimal parameter $a$ is determined by calculating the point of equal cheating probabilities for Alice and Bob $P^A = P^B$ using Eqs. (4) and (5) (see Supplementary Note 3).

For the experimental parameters from Table 1, $P^A = P^B$ is achieved for $a = 0.9$, which is also used in the experiment. The $a$-optimization was performed for each $\{\mu, K\}$ combination, which yielded main Fig. 2.

### Theoretical expectations for projections of prepared states

To calculate the theoretically expected input-output matrix in the main text Fig. 2, we consider the four input states prepared by Alice in the language of the honest protocol in explicit form as a function of $a$ (cf. Eq. (1)):

$$\begin{aligned}
|\phi_{0,0}\rangle &= \sqrt{a}|0\rangle + \sqrt{1-a}|1\rangle \\
|\phi_{0,1}\rangle &= \sqrt{1-a}|0\rangle - \sqrt{a}|1\rangle \\
|\phi_{1,0}\rangle &= \sqrt{a}|0\rangle - \sqrt{1-a}|1\rangle \\
|\phi_{1,1}\rangle &= \sqrt{1-a}|0\rangle + \sqrt{a}|1\rangle.
\end{aligned} \tag{7}$$

Here, the first index denotes the basis and the second the selected state. These states are orthogonal within one basis, normalized, and the bases do not commute. Next, the projections onto the four QCF states, i.e., the overlap of each prepared state with all the other basis states, can be calculated as a function of $a$ (cf. Supplementary Table 2). This was finally used to calculate the matrix illustrating the theoretical expectations in main Fig. 5c for $a = 0.9$ and dividing the projection probabilities by 2 to account for the fact that each basis is chosen in the Bob module with a probability of 50 %.

### Estimation of maximum coin flip rate

The maximum possible rate of secure coin flips is calculated as $r = \frac{1}{T \cdot K_{min}}$, with $T = 12.5$ ns the temporal separation between single-photon pulses and $K_{min}$ the smallest possible value still yielding a quantum advantage for a specific loss. The detection error is modeled to increase with loss as $e = \frac{\eta_{det} e_0 \alpha + 0.5 P_{dc}}{P_{dc} + \eta_{det} t}$ with the attenuation $\alpha$, the dark count rate $P_{dc} = 4 \cdot 10^{-7}$, zero-attenuation error $e_0 = 0.028$, and the detection efficiency $\eta_{det} = 0.425$.

### Dynamic polarization-state encoding

The dynamic polarization-state switching is implemented using a self-built arbitrary waveform generator (AWG) in combination with a fiber-based electro-optic modulator (EOM) in a single-pass configuration. The EOM used in this work is a customized phase-modulator (by EOSPACE Inc.). The AWG driving the EOM is based on a 16-bit digital-to-analog converter (DAC) (by Analog Electronics) driven by an FPGA development board (by AMD). Using the FPGA, a $K \times 2$ ROM structure is designed to store a 2-bit random number sequence of length $K$ using the excitation laser trigger output (Sync) as the main clock signal. For each rising edge of the Sync signal, the FPGA reads the 2-bit value in the

ROM structure and correspondingly updates the 16-bit digital input value for the DAC board, using a low-voltage differential signaling (LVDS) protocol via an FPGA Mezzanine Card connector on the FPGA board. A copy of the Sync signal is generated by the FPGA and provided to the DAC board for reading the digital input pattern and updating the output voltage level. For reading the next entry, the FPGA increases the address input for the ROM by plus one. Once the address value reaches $K$, its value is reset to zero for repeating the pre-stored random modulation pattern. In addition, the FPGA generates a trigger pulse at the beginning of each coin flipping sequence, i.e., once the address value is reset to zero, which is sent to the time tagger input for keeping track of the starting time of each coin flipping attempt. To vary the applied voltage levels while performing calibration routines in real-time, a universal asynchronous receiver-transmitter (UART) module is implemented in the FPGA for sending commands and digital values for each voltage level from our PC to the FPGA using the UART protocol.

### Optimizing modulation voltage levels

To determine the exact voltage levels to be applied to the EOM for preparing the four QSCF states, we follow the calibration routine described below. For this purpose, we use Bob's calibrated polarization-state analyzer described further below. Initially, a direct analytical mapping is performed from the desired qubit states on the Bloch sphere (determined by $a$) to the corresponding Stokes vector $\vec{s}$ on the Poincaré sphere coupled to the quantum channel by Alice (see Supplementary Note 5). Next, we prepare laser pulses tuned to the same wavelength as our single-photon source in the respective polarization states using a polarimeter and optimize the voltage levels step-wise.

In the first step, a periodic sequence of two different voltage levels is generated with the AWG and applied to the EOM for determining the voltage $V_\pi$ necessary to switch between two arbitrary orthogonal polarization states, i.e., $\pi$-rotation on the Poincaré sphere. Using a fiber-based polarization controller, the two prepared polarization states are rotated to match the detection basis. The achievable extinction ratio is monitored by comparing the detected counts projected into two orthogonal basis states. Applying an adjustable electronic temporal delay to the AWG output, we ensure that the single photons arrive while the voltage supplied to the EOM is settled at its target value. This procedure is repeated iteratively for different voltage differences between the two states in the sequence to minimize the QBER. For our experimental setup, the FPGA output voltage yielding the lowest QBER amounts to 330 mV, which, after amplification, corresponds to $V_\pi \approx 3$ V at the EOM.

In a second step, the periodic two-level sequence used above is extended to a four-level switching sequence. While the four BB84-states typically used in QKD are eigenstates of the two non-orthogonal bases, rectilinear and diagonal, the QSCF states are slightly tilted according to the parameter $a$, resulting in a larger overlap between the two bases. For this reason, the two additional voltage levels are not simply shifted by $V_\pi/2$ with respect to the 2-state switching as in the BB84 case. Instead, the correct voltage levels are found by applying a variable shift between the two pairs of voltage levels to minimize the QBER while maintaining the $V_\pi$ difference between orthogonal levels. In each iteration, the four generated states are rotated into the detection reference frame of Bob using the fiber-based polarization controller.

After calibration with the laser, we use linearly polarized single photon pulses to optimize the voltage levels for maximal discrimination between the four known protocol states in the corresponding detection channels. Following the procedure described above, the QBER, defined as the ratio of false detection events in one basis divided by all detection events within this basis, reaches values below 2.0% for a periodic 4-state sequence.

### Four-state Manchester coding

When directly encoding a random polarization state sequence using the voltage levels determined above with our AWG, small time-dependent deviations between the measured and the targeted voltage levels are observed, which are caused by drifts in the DAC stage whenever repeated voltage levels are prepared (which always happens with finite probability in random sequences). This drift results in increased quantum bit errors during the transmission of random polarization-state switching sequences, while it is absent when applying periodic switching sequences as used during the calibration procedure described above. To mitigate this voltage level drift, an advanced coding scheme was employed, which builds on the idea of binary phase-keying, also known as Manchester coding. Here, the electronic clock-rate inside the AWG is doubled from 80 MHz to 160 MHz, and for each sent state, the voltage level corresponding to the orthogonal state is applied. As the optical clock of the excitation laser remains at 80 MHz the polarization-state preparation of the photons can be timed such that polarization switching occurs during the first 6.25 ns of each 12.5 ns laser repetition period (cf. gray shaded window in main Fig. 4b), followed by the orthogonal state's voltage level. This results in an average voltage level inside the control electronics of zero, which effectively suppresses voltage level drifts (see Supplementary Note 6, including Supplementary Fig. S5). While this requires more precise delay control, the previous drift is avoided, allowing for random polarization switching.

### Calibration of qubit state detection

Qubit-state detection is performed on Bob's side using a polarization-state analyzer consisting of a 50:50 beam splitter (BS) passively realizing the random basis choice, followed by two polarizing beam splitters (PBS) onto which the photon' polarization state is rotated using two sets of a half- and a quarter-wave plate, enabling projections into the four protocol states. A high extinction-ratio linear polarizer in each reflected beam path enables a high overall qubit-state discrimination (see further below). The polarization-state analyzer has an overall transmission of $\eta_{Bob} = 50\%$ (excluding the detectors). To optimize the discrimination of the four QSCF states in the Bob module, a polarimeter in combination with a linear polarizer is used in the four output ports, while carefully adjusting the wave-plates. Applying this calibration procedure, we achieve a contribution of the Bob module to the overall QBER below 0.1%. Photons from the four outputs are finally detected via a superconducting nanowire single photon detector (Single Quantum) with $\eta_{Det} = 85\%$ detection efficiency. Not least, residual temporal delays between the four detection channels are carefully compensated by matching the arrival times of laser pulses tuned to the emission wavelength of our single-photon source.

### Data analysis

The first step to implement the full coin flipping protocol and for generating secure random bits is to synchronize the prepared states of Alice with the detected states at Bob. Alice has a list of the $K$ states she prepared during the sequence, while Bob only detects a single photon per sequence on average. Summing over many realizations of the $K$-long pulse sequence allows Bob to guess the state he most likely received in each time bin of the sequence (cf. Supplementary Fig. S5a). The resulting list (or a set) of detected states [1, 2, 3, 4] can be correlated with Alice' list to find the relative temporal offset between the two lists. This is done by iteratively computing the average of the element-wise difference while rolling the index of one list. See Supplementary Note 7, including extended data illustrated in Supplementary Fig. S6.

Using the correct temporal offset, as determined above, each coin flip can be evaluated on a single coin flip level following the strong coin flipping protocol introduced in the main text by exchanging random bits, comparing the basis and state and, if the protocol did not abort, generating a secure random bit.

## Data availability

The data generated in this study have been deposited in the Zenodo database under accession code https://zenodo.org/records/18436939.

## Code availability

All codes produced during this research are available from the corresponding authors upon request.

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

## Acknowledgements

The authors gratefully acknowledge early contributions to the experimental methodology and software by Timm Gao, experimental support by Bhavana Panchumarthi, Aodhan Corrigan, and Calista Eitel-Porter, as well as technical support by Johannes Schall, Sven Rodt, Stephan Reitzenstein, and Chengao Yang. The authors further acknowledge financial support by the German Federal Ministry of Research, Technology and Space (BMFTR) via the project "QuSecure" (Grant No. 13N14876) within the funding program Photonic Research Germany, the BMFTR joint projects "tubLAN Q.0" (Grant No. 16KISQ087K) as well as QuNET+ICLink (Grant No. 16KIS1967) in the context of the federal government's research framework in IT-security "Digital. Secure. Sovereign.", and the Einstein Foundation via the Einstein Research Unit "Quantum Devices". A.P. also acknowledges financial support by the German Research Foundation (DFG) via the Emmy Noether (Grant No. 418294583). H.L., S.L., H.N., and Z.N. acknowledge financial support by the Chinese Academy of Sciences Project for Young Scientists in Basic Research (Grant No. YSBR-112), the National Natural Science Foundation of China (Grant No. 12494601), and the Innovation Program for Quantum Science and Technology (Grant No. 2021ZD0300801).

## Author contributions

D.A.V. and K.K. set up the quantum coin flipping experiment under the supervision of M.v.H. and T.H.; F.D. and D.A.V. performed the protocol simulations. L.R. designed and fabricated the single-photon source based on the quantum dot wafer material provided/grown by H.L., S.L., H.N., and Z.N. Furthermore, D.A.V., F.D., A.P., and T.H. prepared the paper with inputs from all authors; A.P. supervised the theoretical and T.H. the experimental aspects of the project; T.H. and A.P. jointly conceived the project.

## Funding

## Competing interests

The authors declare no competing interests.
