## [Transparent Peer Review file · Nature Communications]

Single-Photon Advantage in Quantum Cryptography Beyond QKD

Corresponding Author: Professor Tobias Heindel

Version 0:

Reviewer comments:

Reviewer #1

(Remarks to the Author)

In this work, the Authors identify a security advantage in terms of cheating probability for a well-studied quantum-cryptographic task known as quantum strong coin flipping (QSCF), enabled by the use of single-photon sources (SPS) instead of weak coherent pulses (WCP). Using a quasi-resonantly pumped semiconductor quantum dot emitting at 921 nm with a second-order correlation function at zero time delay of around 3%, the Authors demonstrate this advantage experimentally for several values of channel loss, also maintaining quantum advantage over classical implementations of SCF with a similar abort probability.

The manuscript reads in a clear and structured way. Detailed theoretical and experimental methodologies, along with the experimental results, are presented in the Main, Methods and Supplementary. In principle, I believe that experimentally demonstrating the first advantage of single-photon sources (SPS) over Poissonian alternatives in a beyond-QKD protocol is worth pursuing: it provides a new benchmark and elegant milestone that should be of interest to both single-photon and quantum cryptography communities. I also appreciate the effort taken in actively generating polarization states that lie much closer to each other on the Bloch sphere than standard BB84 states, along with measures taken to limit voltage drifts. However, the demonstrated advantage of SPS over WCS is very small in my opinion (~0.3% absolute difference at 0 dB loss), and the actual cheating probabilities of ~90% are unfortunately too high to provide any valuable secure application. In essence, I feel like the theoretical aspects do not provide any novelty, since the chosen QSCF protocol, its practical security analysis and its associated parameter space optimization (applicable to both SPS and WCS) were extensively studied in Refs. [10,11,13,15]. This would not necessarily be a problem in itself, but the main claim of the manuscript, i.e. that SPS outperform WCS for this protocol, might be overestimated due to the bounds on the cheating probabilities not being tight (as mentioned by the Authors and shown in the security analysis adapted from previous works).

In order to provide a fair or more significant comparison, I would at least suggest some effort on the theory side to make the security bounds tighter, which would in turn make the source comparison more fair, and/or formulate altogether a different figure of merit where the source advantage might be more substantial. For these reasons, I do not recommend publication in Nature Communications at this stage, unless these concerns are addressed. In the points below, I give suggestions that can help the Authors, along with some more minor clarity/definition comments.

Possibilities/limits of SPS advantage:

1) Protocol/quantum advantage: to increase the fairness of the source comparison and novelty of the overall results, I would suggest improving the security analysis (with numerical methods to estimate some additional $A_{i>4}$ in Eq.(4) if this is too cumbersome analytically) to improve the (quantum) advantage over classical methods for a fixed, experimental abort probability. This would additionally make the source comparison between SPS and WCS more fair, as the cheating due to multiphoton noise would be less overestimated. If the difference turns out not to be significant, I am happy with the current analysis.

2) Source advantage: The advantage of SPS over WCS is small (~0.3% at 0 dB loss), and likely a little smaller if the security bounds for the QSCF protocol are tightened (since you would get less penalized by multiphoton emission, which is the main difference between WCS and SPS at a fixed mean photon number). Overall, I am wondering whether, for this very reason, there could not be a more impressive source advantage to demonstrate as a function of losses: for example, over what distance/loss are SPS able to maintain a quantum advantage compared to WCP? I have the feeling that this would be a

more impressive figure of merit, since the multiphoton advantage of SPS would get significant with higher loss. Furthermore, although the employed QD source has some limitations, I would at least suggest plotting an additional curve in Fig.2a that shows how much better an ideal (or the best currently published) QD could perform in principle.

3) Frequency conversion: The employed QD emits at 921nm, but QSCF is a communication protocol and will most likely be implemented at telecom wavelengths when deployed over standard SMF fibers. How do the Authors expect frequency conversion to affect their demonstrated advantage? More precisely, incorporating typical numbers for the losses and multiphoton noise introduced by frequency conversion, would SPS still gain an advantage over WCP for associated values of average photon number " μ " and rounds K ?

Definitions and clarity:

- Line 21, "The most fundamental quantum cryptographic building block [...] is quantum coin flipping": this is not correct per se, as, both in the classical and quantum world, coin flipping may be constructed from much more fundamental primitives like bit commitment and oblivious transfer. Please attenuate this claim by changing it (for example) to "one of the fundamental building blocks...".
- Line 43, "eg: for doing business": this does not provide the Reader with any information regarding the applications of coin flipping. Could the Authors cite one or two relevant use-cases?
- Line 120, "under the fairness constraint": this quantity/constraint has not been defined before. To make things clear for the uninformed Reader, I would strongly recommend that the Authors introduce all the relevant properties of quantum coin flipping earlier in the draft, for example before or after the protocol description. These should at least include fairness (probability of winning the coin flip should be equal for honest parties), balance (upper-bounds on the cheating probabilities should be equal for both parties), and abort probability.
- Line 170, "results in the emission spectrum in Figure 2b": this should read "Figure 3b".

Reviewer #2

(Remarks to the Author)

In this article, the authors implemented a quantum strong coin flipping protocol, which is an interesting cryptographic primitive useful for sharing randomness without trust, as well as serving as a subroutine for more complex cryptographic protocols like commitment schemes. The main difference and novel analysis compared to the previously proposed protocol "Experimental Plug and Play Quantum Coin Flipping" is that they replaced the coherent light source with a high-fidelity deterministic single photon source. They analyzed the new advantages offered by using a single photon source in this protocol and observed a decrease in the cheating probability compared to the weak coherent pulse model previously proposed. Additionally, they explored the effects of losses on the quantum advantage observed, providing a more comprehensive analysis.

The authors presented both simulated and implemented data that are consistent and easy to understand. For the experimental setup, they employed a cutting-edge photon source recently developed by some of the authors, along with dynamic state encoding controlled by FPGAs, which has been refined to reduce error levels.

Overall, the article offers new insights and analysis, as well as a well-developed experimental setup. However, I have one suggestion for improvement: In the abstract, the authors mention that probabilistic photon sources, which I assumed to be SPDC sources, suffer from "fundamental limitations," but this point is not discussed further or referenced afterward. Properly referencing and detailing these limitations would strengthen the argument more.

Reviewer #3

(Remarks to the Author)

In the work "Single-photon advantage in quantum cryptography beyond QKD," the authors present the implementation of a quantum strong coin-flipping protocol using a deterministic single-photon source based on a semiconductor quantum dot. They demonstrate, first through simulations and subsequently in experiments, not only the advantage of the quantum protocol over its classical counterpart, but also the benefit of implementing single-photon states compared to coherent states.

I find the paper both interesting and technically sound. However, before making a final decision, I would like the authors to address the following comments and questions:

- I believe the title does not accurately reflect the content of the work. The phrase "in quantum cryptography beyond QKD" suggests a broad survey of all quantum cryptographic applications (excluding QKD) where single photons offer an advantage. Since this work focuses on a specific protocol, it might be clearer and more appropriate to reference that directly in the title.
- In the first section, while describing the protocol, some points could be clarified. For example, in point 2, Bob chooses a basis β_i , but it is unclear what basis this corresponds to without referring to the figure. In point 3, it is stated that Bob picks a random number b_j , but it is not specified that this should be a binary value.
- The protocol appears to describe only a single round of the experiment. How is parameter estimation performed in practice? Is there an abort condition depending on the quantum bit error rate (QBER)?
- In the experimental section, the authors test the setup in a free-space quantum channel with varying levels of loss and

compare it to the expected performance in fiber. However, why was the experiment not performed directly using optical fiber? Are there challenges related to coupling into the fiber?

- The authors highlight the advantage of single-photon sources in terms of a reduced cheating probability. However, the reported difference is around 0.3%, which seems quite small. Considering the significantly easier implementation using coherent states, does this marginal advantage still justify switching to a single-photon source?

Version 1:

Reviewer comments:

Reviewer #1

(Remarks to the Author)

I would like to thank the Authors for addressing my comments in a very detailed and convincing manner (especially the study of the impact of terms $A_{i>4}$ and the addition of ideal SPS curves to show the full expected potential of SPS), and for implementing the changes that make the article more informative.

I believe that the manuscript is now suitable for publication, provided that the following minor comment is implemented:

- I really appreciate the addition of Figure 7, which displays an advantage of SPS over WCP for achievable distance that is in my opinion more impressive than the pure (small) gain in terms of bias at zero distance. In the figure title however, could you please change "unbiased" coin flips to "secure" coin flips? A Reader might get confused otherwise, since the coin flips DO have a finite bias.

Reviewer #3

(Remarks to the Author)

I thank the authors for addressing my comments in a precise and thorough manner. I acknowledge and accept the authors' decision to retain their original choice of title, which is well within their rights. I particularly appreciate the honest and thoughtful response to my final comment concerning the academic relevance of the reported results.

For these reasons, I consider the manuscript worthy of publication in Nature Communications.

Response Letter to Nature Communications
Manuscript NCOMMS-25-16152-T
by Daniel A. Vajner et al.

December 11, 2025

We thank the Reviewers for their assessment of our work and the, in principle, very positive feedback. The constructive criticism helped us to further improve our manuscript. We carefully read all remarks and addressed the points raised as detailed in the following. Here, we mark reviewer comments in **blue**, our responses in black, and revised/new text in **red**. The detailed point-by-point response to the Reviewers below is complemented by a marked-up version of the Revised Manuscript and the Supplementary Information file, highlighting the changes made.

Tobias Heindel
on behalf of all authors

Reviewer 1

In this work, the Authors identify a security advantage in terms of cheating probability for a well-studied quantum-cryptographic task known as quantum strong coin flipping (QSCF), enabled by the use of single-photon sources (SPS) instead of weak coherent pulses (WCP). Using a quasi-resonantly pumped semiconductor quantum dot emitting at 921 nm with a second-order correlation function at zero time delay of around 3%, the Authors demonstrate this advantage experimentally for several values of channel loss, also maintaining quantum advantage over classical implementations of SCF with a similar abort probability.

The manuscript reads in a clear and structured way. Detailed theoretical and experimental methodologies, along with the experimental results, are presented in the Main, Methods and Supplementary. In principle, I believe that experimentally demonstrating the first advantage of single-photon sources (SPS) over Poissonian alternatives in a beyond-QKD protocol is worth pursuing: it provides a new benchmark and elegant milestone that should be of interest to both single-photon and quantum cryptography communities. I also appreciate the effort taken in actively generating polarization states that lie much closer to each other on the Bloch sphere than standard BB84 states, along with measures taken to limit voltage drifts. [...]

We thank the reviewer for assessing our manuscript and the, in principle, very positive feedback to our work.

However, the demonstrated advantage of SPS over WCS is very small in my opinion (0.3% absolute difference at 0 dB loss), and the actual cheating probabilities of 90% are unfortunately too high to provide any valuable secure application. In essence, I feel like the theoretical aspects do not provide any novelty, since the chosen QSCF protocol, its practical security analysis and its associated parameter space optimization (applicable to both SPS and WCS) were extensively studied in Refs. [10,11,13,15]. This would not necessarily be a problem in itself, but the main claim of the manuscript, i.e. that SPS outperform WCS for this protocol, might be overestimated due to the bounds on the cheating probabilities not being tight (as mentioned by the Authors and shown in the security analysis adapted from previous works).

Doubtlessly, a larger quantum advantage and lower cheating probabilities would be desirable. We would like to stress, however, that our work reports the first-ever implementation of quantum coin flipping using single-photon states and we believe that demonstrating an advantage compared to probabilistic sources, although small, is a quite remarkable achievement in a first step. Concerning the relatively high cheating probability, we need to emphasize that the specific protocol we chose is a quantum strong coin flipping (QSCF) protocol that considers cheating in both directions of the coin flip, as well as realistic imperfections, resulting in cheating probabilities that cannot be significantly smaller than $\approx 89\%$. Using different coin flipping protocols with added assumptions, e.g. an adversary with a noisy storage, the cheating probabilities can be reduced further. One should keep in mind that a perfect quantum coin flipping protocol, unlike quantum key distribution, does not exist. Finally, when we compare to the classical protocol, its cheating probability is below 100% only because we assume a finite honest abort probability. In principle classical errors can be corrected and bright pulses can be used that can be amplified within the channel, so that a negligible honest abort probability would make classical cheating always possible, rendering the reduction to 90% a significant improvement. In this context we also refer to our response to the last comment by Reviewer #3 as well as the text we added to the "Conclusion and Outlook" section of our revised manuscript, discussing perspectives of future work on the protocol side.

In addition, we would like to stress that, while the implemented QSCF protocol has indeed been studied in previous work, an in-depth discussion and simulation of the possible single-photon advantage as presented in our manuscript has not been explored in prior art. We believe that this also contributes significantly to the high degree of novelty of our work, also beyond the experimental demonstration itself. Finding tighter bounds within the framework of a refined security analysis remains an interesting route to take. However, as we discuss below, a refined analysis of Bob's cheating probability would not lead to significant changes for the chosen parameter range.

In order to provide a fair or more significant comparison, I would at least suggest some effort on the theory side to make the security bounds tighter, which

would in turn make the source comparison more fair, and/or formulate altogether a different figure of merit where the source advantage might be more substantial. For these reasons, I do not recommend publication in Nature Communications at this stage, unless these concerns are addressed. In the points below, I give suggestions that can help the Authors, along with some more minor clarity/definition comments.

We thank the reviewer for their opinion on these aspects and the suggestions how to address them. We considered the suggestions as detailed in the following.

Possibilities/limits of SPS advantage:

1) Protocol/quantum advantage: to increase the fairness of the source comparison and novelty of the overall results, I would suggest improving the security analysis (with numerical methods to estimate some additional $A_i > 4$ in Eq.(4) if this is too cumbersome analytically) to improve the (quantum) advantage over classical methods for a fixed, experimental abort probability. This would additionally make the source comparison between SPS and WCS more fair, as the cheating due to multiphoton noise would be less overestimated. If the difference turns out not to be significant, I am happy with the current analysis.

We appreciate the reviewer’s suggestion to take a closer look at the fairness of our comparison. For this purpose, we first revisit the meaning of the terms A_1 to A_4 in Eq.(4), before analyzing the impact of the terms $A_i > 4$.

The security analysis in our work covers the following four cases:

- A_1 : Bob does not detect a single photon
- A_2 : Bob detects at least one 1-photon pulse
- A_3 : Bob detects exactly one 2-photon pulse
- A_4 : Bob detects exactly one 2-photon pulse and at least one 1-photon pulse

Higher-order terms, describing one 3-photon pulse or two 2-photon pulses, are not only less likely, but also their cheating probability is very close to the conservatively assumed 100% (if not explicitly quantified). In our work, we decided to focus on a parameter range, in which the protocol can be carried out at reasonable rates. Choosing a very high value for K with respect to $1/\mu$ will always allow cheating both for a laser and a QD source, leading to a vanishing quantum advantage. Choosing a very small K will make cheating very rare, but the probability of detecting a photon will also be very small, which means the coin flipping protocol will abort in most cases. In the typical regime of $K \approx 1/\mu$, the probability of all additional cases $A_{i>4}$ is $< 10^{-8}$ for attenuated laser pulses and $< 10^{-12}$ for our QD light source (see Response-Figure 1).

Hence, we deem the conservative assumption of attributing a probability of 1 to cheat in this case sufficient. Even in a more precise calculation, this

Response-Figure 1: Calculated probability of the additional explicit cheating cases of Bob as $P(A_{i>4}) = 1 - \sum_{i \leq 4} P(A_i)$ both for the QD single photon source of $g^{(2)}(0) = 0.03$ (solid line) and an attenuated laser source (WCP, dashed line).

value would remain in the range between 0.98 (cheating probability in case A_4) and 1 (maximum possible probability), which is a small difference when being compared to 10^{-12} . We conclude that when explicitly adding additional terms $A_{i>4}$, the quantum cheating probability will not be significantly reduced compared to the classical one in the relevant parameter range. Rephrasing our answer, we agree with the reviewer that the WCP cheating probability would be slightly more reduced by including these additional cases, but as shown above, the absolute values are insignificant in the relevant parameter range.

We modified the Methods section "Analytical expressions for cheating probabilities" of the revised manuscript as follows considering the discussion from above:

*For Bob, the optimum cheating strategy depends on whether he has detected 0, 1, or 2 photons, assuming that he can differentiate the photon number. **These cases are (cf. Ref. [1]):***

- (A_1) Bob does not detect a single photon
- (A_2) Bob detects at least one 1-photon pulse
- (A_3) Bob detects exactly one 2-photon pulse
- (A_4) Bob detects exactly one 2-photon and at least one 1-photon pulse

Summing the contributions leads to an upper bound for Bob's total cheating probability of [...]

For simplicity, Bob is assumed to be able to cheat with unity probability in all other cases $A_{i>4}$. Hence, the cheating probability bound defined for Bob above is

not tight and lower bounds are possible in principle. Note, however, while more explicit events could be considered to further tighten Bob's bound, the probability for higher-order multi-photon events is not only very low, but also their cheating probability is very close to the conservatively assumed 100%. In the reasonable parameter regime of $K \approx 1/\mu$ used in our work, which allows for practical coin flipping rates, the probability of all additional cases $A_{i>4}$ can be estimated to be $< 10^{-8}$ for attenuated laser pulses and $< 10^{-12}$ for our QD light source. Hence, we consider the impact of terms $A_{i>4}$ negligible for our study.

2) Source advantage: The advantage of SPS over WCS is small ($\approx 0.3\%$ at 0 dB loss), and likely a little smaller if the security bounds for the QSCF protocol are tightened (since you would get less penalized by multi-photon emission, which is the main difference between WCS and SPS at a fixed mean photon number). Overall, I am wondering whether, for this very reason, there could not be a more impressive source advantage to demonstrate as a function of losses: for example, over what distance/loss are SPS able to maintain a quantum advantage compared to WCP? I have the feeling that this would be a more impressive figure of merit, since the multi-photon advantage of SPS would get significant with higher loss. Furthermore, although the employed QD source has some limitations, I would at least suggest plotting an additional curve in Fig.2a that shows how much better an ideal (or the best currently published) QD could perform in principle.

We thank the reviewer for this comment. We will first discuss quantum and single photon advantage separately.

Concerning the quantum advantage over an equivalent classical protocol, the main limitation here is the protocol choice. For realistic parameters, a will always be around 0.9. Hence, in the regime in which Bob receives on average one photon per round, even if no multi-photon events exist, he will also be able to cheat with about 90% probability. In this case, the quantum advantage depends solely on the equivalent classical cheating probability, which is set by the honest abort probability as a function of the QBER. With sufficient computational power I can always cheat classically; however, a non-zero bit error leads to honest aborts, and hence cheating is not possible in these cases. Thus, the quantum advantage would be greatest for $e = 0$ and is limited by the QBER of 2.8% here.

Concerning the single photon advantage between QD source and attenuated laser, the striking difference lies not in the absolute number of the cheating probabilities, but in the much larger parameter range in which a quantum advantage can be maintained. This is illustrated in Figure 2 (b,c) of our manuscript. Note that here we already optimize the parameter a to achieve fairness and minimize the cheating probability for each combination of μ and K .

Concerning the absolute difference in cheating probabilities, one has to face a trade-off. As seen in Figure 2(a) of the manuscript, one cannot simultaneously maximize the difference between quantum vs. classical and WCP vs. SPS. The

Response-Figure 2: Difference between classical and SPS cheating probability (green line), classical and WCP (blue line) and WCP and SPS (red line) to see that single photon advantage and quantum advantage do not happen for the same K for fixed $\mu = 0.0013$. The maximum quantum advantage is marked by the green vertical line, while the maximum single photon advantage is marked by the red vertical line.

maximum difference between the laser and QD cheating probability occurs for parameters that no longer correspond to the maximum quantum advantage, as can be seen more clearly in Response-Figure 2 showing the differences in the respective cheating probabilities. One can see how, while the SPS has a quantum advantage for a larger parameter range than the WCP source (green vs. blue curve), the red SPS is also always better than the WCP source (red curve always larger than zero). But more importantly, the maximum of the green curve (maximum quantum advantage) occurs for parameters different from the maximum of the red curve (maximum single photon advantage). In the manuscript so far, we chose parameters that clearly yield a quantum advantage, while showing that the single photon advantage is present as well. Additionally, choosing a smaller K means that the sequences are shorter and more coin flips can be performed per time. We included this discussion and Fig. 2 in the revised Supplementary Material, Note 3, and added the following text to the revised discussion of Fig. 2 in the main text:

Note, that the advantage of our SPS compared to WCPs can exceed 8% for larger K (see **Supplementary Note 3**). This, however, occurs in a regime with a reduced advantage compared to the classical case. In our work, we therefore chose parameters that clearly yield a quantum advantage while simultaneously enabling a single-photon advantage.

Therefore, in the parameter range with significant quantum advantage using an ideal single photon source (without multi-photon pulses) would not signifi-

Response-Figure 3: Updated Figure 2 of the manuscript, including a perfect SPS in panel (a)

cantly reduce the cheating probability further. However, an ideal single photon source would be able to maintain that low cheating probability for arbitrarily high mean photon numbers (or protocol rounds). To underline this, we have added a line for an ideal SPS in Figure 2(a) of the manuscript, as suggested (cf. Response-Figure 3. When increasing K , a perfect single photon source does not have more multi-photon events, thus the cheating probability does not increase. We have added the following note to the manuscript:

“The dashed green line indicates an ideal SPS without any multi-photon pulses ($g^{(2)}(0) = 0$). In this case, no 2-photon events can be used to cheat, thus preserving the quantum advantage for large K . However, for $K \ll 1/\mu$ the probability of multi-photon events becomes negligible even for a non-ideal single-photon sources, which is why the maximum quantum advantage is the same, and even an ideal single-photon source would not further reduce the cheating probability for this protocol and experimental parameters.”

Finally, the main advantage of a single photon source in QSCF being the larger parameter range is less pronounced in the experiment because we can not freely choose the parameters. When introducing loss, the quantum advantage decreases due to 2 reasons:

- Bob receives fewer photons, but keeps K the same, so the honest abort probability will increase as more rounds end without any photon detections, causing the classical cheating probability to go down, which kills the quantum advantage. This is why each curve in Figure 6(b) of the manuscript goes down continuously as the loss increases.
- Even if Bob increases K to keep the honest abort probability the same for higher loss, he will collect more dark count events in each round, increasing the QBER, which again causes the classical cheating probability to go down. This is why different curves for different QBER in Figure 6(b) are shifted with respect to each other.

Nevertheless, when Alice and Bob increase K to compensate for the loss, the occasional multi-photon events become more relevant and the single

Response-Figure 4: Updated Figure 6 from the manuscript now including also the attenuated laser source as a reference dashed line in panel (b)

photon advantage slightly more pronounced, which is why we have updated Figure 6b of the manuscript (c.f. Response-Figure 4) with a version that contains also an attenuated source of the same mean photon number in form of the dashed lines.

As the main difference in terms of loss tolerance is that an ideal single photon source can be operated at a much higher μ or K in the first place, (without significant multi-photon events, allowing it to tolerate more loss along the way and higher K), we added an additional Figure 7 in the manuscript in which we extrapolate the performance for a brighter single photon source of $\mu = 0.5$ and less multi-photon events with $g^{(2)}(0) = 0.001$, which has been achieved by different groups already (refs.). Additionally, to make the advantage more clear, we estimate an upper bound for the maximum possible rate of unbiased coin flips, calculated as $r = \frac{1}{T \cdot K}$, in which $T = 12.5$ ns is the time period between two excitation pulses and K is the smallest possible value that still yields a quantum advantage for a certain loss. The detection error is modeled to increase with loss as $e = \frac{\eta_{Det} e_0 t + 0.5 P_{dc}}{P_{dc} + \eta_{Det} t}$ with the attenuation t , the dark count rate $P_{dc} = 4 \cdot 10^{-7}$, zero-attenuation error $e_0 = 0.028$ and the detection efficiency $\eta_{Det} = 0.425$. Being able to operate at a larger mean photon number, allows choosing a shorter protocol sequence K , so that more coin flip rounds are possible per time (higher rate). Additionally, being able to tolerate a larger K due to less multi-photon

Response-Figure 5: Additional Figure 7 in the manuscript showing the maximally possible rate of unbiased coin flips for the QD source used here (green line), an equivalent attenuated pulse source (blue line), and an improved single photon source (red line).

events means that coin flips are still possible for a higher loss (longer range), as shown in Response-Figure 5.

Note that in the experiment we maximized the quantum advantage and not the coin flip rate, which required a larger K and thus the observed coin flip rate in the experiment was smaller. Here, we show the maximum possible rate while preserving a small quantum advantage. The following manuscript text was added.

”To further explore the effect of loss, Figure 5 shows the maximum possible rate of unbiased coin flips R , which is calculated from the system clock-rate (80 MHz) divided by the smallest possible value of K still yielding a positive quantum advantage. Here, we assumed that the error at zero attenuation is $e_0 = 2.8\%$ and modeled the increase in error with loss (see Methods). By comparing our single-photon source (green line) to the equivalent attenuated laser source (blue line), we confirm that the sub-Poissonian photon statistics lead to performance improvements. Assuming an improved implementation (red line), using a single-photon source combining the best values for $\mu = 0.28$ [2] and $g^{(2)}(0) = 0.0001$ [3] achieved with quantum dots to date, would lead to substantial performance improvements in single-photon QSCF. Here, the higher source efficiency allows for shorter protocol sequences K , which in turn enable higher coin flipping rates, while larger K can be tolerated at higher loss.”

3) Frequency conversion: The employed QD emits at 921 nm, but QSCF is a communication protocol and will most likely be implemented at telecom wavelengths when deployed over standard SMF fibers. How do the Authors expect frequency conversion to affect their demonstrated advantage? More precisely, incorporating typical numbers for the losses and multi-photon noise introduced by frequency conversion, would SPS still gain an advantage over WCP for associated values of average photon number “ μ ” and rounds K ?

We thank the reviewer for the suggestion to explore how quantum frequency conversion (QFC) would affect the performance of our implementation. In the context of quantum cryptography in point-to-point or prepare-and-measure settings, QFC can be considered a useful transition technology [4]. In fact, pioneering early work demonstrated already, that QFC maintains the quantum optical properties of QD sources to a very large extent [5], and additional multi-photon noise can be mostly avoided [6]. However, the additional losses of the QFC stage on Alice’s side (typically ≈ 3 dB) lower the achievable μ -values, which reduces the maximum tolerable loss.

From Figure 2 of the manuscript it becomes evident that, without additional loss, also a reduction of a factor of 2 of the mean photon number due to frequency conversion efficiency could be compensated by a larger K and a quantum advantage is still possible. Concerning the loss dependence, Figure 6 of the manuscript is general as it uses loss in dB, but of course when moving from 930 nm to 1550 nm in optical glass fibers, the same loss in dB would correspond to about 10x longer fiber distance.

As a side-note, more interesting use-cases of quantum frequency conversion are schemes relying on two-photon interference from remote quantum emitters (e.g., MDI-QKD, quantum teleportation, quantum repeaters), where the additional degree of freedom for spectral tuning via the pump laser frequency can be exploited to spectrally match slightly distinguishable quantum light sources [7].

For the reasons stated above, QD sources directly emitting in the telecom C-band are beneficial for quantum coin flipping, especially in terms of the achievable distance in standard optical fibers. Looking at the recent advances in this context (see e.g., review by Holewa et al. [8]), we anticipate telecom C-band compatible QD sources soon to match the performance of the best shorter wavelength counterparts.

We have added the following to the discussion section of the manuscript:

Moreover, we conducted QSCF experiments under variable attenuation inside the quantum channel with different fixed values of K , revealing that a quantum advantage can be maintained up to 3 dB of additional loss. Calculations of the maximum achievable coin flipping rate show that a quantum advantage of up to 15 dB loss is possible in principle for our experimental parameters using optimized values of K . Using single-photon sources operating in the telecom C-band

(or lower wavelength emitters using frequency conversion) in future experiments will enable unbiased quantum coin flipping over distances of several tens of kilometers.

Definitions and clarity:

- Line 21, "The most fundamental quantum cryptographic building block [...] is quantum coin flipping": this is not correct per se, as, both in the classical and quantum world, coin flipping may be constructed from much more fundamental primitives like bit commitment and oblivious transfer. Please attenuate this claim by changing it (for example) to "one of the fundamental building blocks...".

We thank the reviewer for this comment and changed the wording accordingly:
One of the fundamental quantum cryptographic building blocks [...]

- Line 43, "eg: for doing business": this does not provide the Reader with any information regarding the applications of coin flipping. Could the Authors cite one or two relevant use-cases?

We agree with the reviewer that this phrase was too unspecific and revised the sentence accordingly:

Many practical use-cases, including randomized leader election, commitment schemes, multi-party computation, or online casinos, involve two or more parties who do not know or trust each other, but still want to interact [9, 10].

- Line 120, "under the fairness constraint": this quantity/constraint has not been defined before. To make things clear for the uninformed Reader, I would strongly recommend that the Authors introduce all the relevant properties of quantum coin flipping earlier in the draft, for example before or after the protocol description. These should at least include fairness (probability of winning the coin flip should be equal for honest parties), balance (upper-bounds on the cheating probabilities should be equal for both parties), and abort probability.

We thank the reviewer for this suggestion and added the following definitions to the section introducing the QSCF protocol:

The important figures of merit for a coin flipping protocol are the honest winning probabilities $P_h^{(A)}$ and $P_h^{(B)}$ of Alice and Bob, their dishonest cheating probabilities P^A and P^B , as well as the honest abort probability P_{AB} , which describes protocol aborts due to photon loss or errors without cheating. In this context, a protocol is called balanced if Alice and Bob have the same honest winning probabilities ($P_h^{(A)} = P_h^{(B)}$), while fairness implies equal dishonest cheating probabilities ($P^A = P^B$).

- Line 170, "results in the emission spectrum in Figure 2b": this should read

“Figure 3b”.

The authors thank the reviewer for thorough reading and corrected the typo.

Reviewer 2

In this article, the authors implemented a quantum strong coin flipping protocol, which is an interesting cryptographic primitive useful for sharing randomness without trust, as well as serving as a subroutine for more complex cryptographic protocols like commitment schemes. The main difference and novel analysis compared to the previously proposed protocol “Experimental Plug and Play Quantum Coin Flipping” is that they replaced the coherent light source with a high-fidelity deterministic single photon source. They analyzed the new advantages offered by using a single photon source in this protocol and observed a decrease in the cheating probability compared to the weak coherent pulse model previously proposed. Additionally, they explored the effects of losses on the quantum advantage observed, providing a more comprehensive analysis.

The authors presented both simulated and implemented data that are consistent and easy to understand. For the experimental setup, they employed a cutting-edge photon source recently developed by some of the authors, along with dynamic state encoding controlled by FPGAs, which has been refined to reduce error levels.

Overall, the article offers new insights and analysis, as well as a well-developed experimental setup. [...]

The authors thank the reviewer for the positive assessment of our work.

However, I have one suggestion for improvement: In the abstract, the authors mention that probabilistic photon sources, which I assumed to be SPDC sources, suffer from “fundamental limitations,” but this point is not discussed further or referenced afterward. Properly referencing and detailing these limitations would strengthen the argument more.

We thank the reviewer for this comment. Here, we refer to both, phase-randomized weak coherent pulses (attenuated laser pulses) and SPDC sources. Both have a photon number distribution that differs from an ideal single-photon source, making multi-pulse events more likely. These give additional cheating possibilities to Bob which reduces the quantum advantage. In the manuscript, we discuss this difference when comparing, for example, the parameter range in which a quantum advantage is present between a realistic single-photon source and an attenuated laser source in Figure 2. To make this point clearer also in the abstract, we have rephrased the sentence as follows:

*[...] quantum coin flipping, which has been **investigated** only in few experimental studies to date, all of which used probabilistic quantum light sources. **With these types of light sources, the unavoidable multi-photon contributions increase the probability of cheating, and thus impose fundamental limitations.***

Reviewer 3

In the work “Single-photon advantage in quantum cryptography beyond QKD,” the authors present the implementation of a quantum strong coin-flipping protocol using a deterministic single-photon source based on a semiconductor quantum dot. They demonstrate, first through simulations and subsequently in experiments, not only the advantage of the quantum protocol over its classical counterpart, but also the benefit of implementing single-photon states compared to coherent states.

I find the paper both interesting and technically sound. [...]

We thank the reviewer for the positive assessment of our work.

However, before making a final decision, I would like the authors to address the following comments and questions:

- I believe the title does not accurately reflect the content of the work. The phrase “in quantum cryptography beyond QKD” suggests a broad survey of all quantum cryptographic applications (excluding QKD) where single photons offer an advantage. Since this work focuses on a specific protocol, it might be clearer and more appropriate to reference that directly in the title.

We agree with the reviewer, that a more specific title might be clearer for the expert reader. In case of our work, however, which is the first to report a single-photon advantage in an experimental realization of a cryptographic primitive beyond QKD, we believe that our original title increases the accessibility of this work, especially for scientists outside the field of quantum information. Also, in the field of QKD many reports do not specify the exact protocol (e.g. BB84, BBM92, etc.) used in their work, which is then specified further in the abstract - as we do. For this reason, we prefer to keep the title as is and hope the reviewer is able to accept this choice.

The protocol appears to describe only a single round of the experiment. How is parameter estimation performed in practice? Is there an abort condition depending on the quantum bit error rate (QBER)?

We thank the reviewer for these questions.

A single round of the coin flipping protocol, which consists of the exchange of K pulses, with K chosen such that on average 1 photon is detected, leads to the exchange of a single random bit between Alice and Bob. Hence, within a single round, no meaningful statistics can be collected to estimate the QBER. A round is aborted whenever nothing is received on Bobs' side or if the final comparison step fails. However, the QBER value does not directly enter the post-processing. For instance, in standard BB84 QKD, a subset of the sifted key is used to statistically estimate a QBER in the parameter estimation step

and a QBER of more than 11% cannot be compensated anymore.

Here, to perform quantum coin flipping the QBER does not need to be calculated explicitly in the experiment. However, higher QBER makes an honest abort more likely, which means that more rounds are required for the same number of random bits.

Still the QBER is required to quantify the quantum advantage, as the equivalent classical cheating probability depends on the honest abort probability (and thus on the QBER and loss). At a very high QBER the protocol aborts most of the time, making classical cheating difficult, and no quantum advantage remains. Thus we estimate the QBER in our work by averaging over multiple rounds, which facilitates the comparison to the classical case. This is not required during the protocol execution. To quantify the quantum cheating probability, the parameter a and the photon statistics are sufficient.

Interestingly, subsequent protocol rounds are not only independent in the experiment, combining them does not provide an advantage. The cheating probability cannot be further reduced by applying a hash function to the results of multiple rounds [11], so the rounds must be treated separately. In principle, the QBER could change from round to round, resulting in different abort probabilities, which would in turn reduce the quantum advantage of some rounds - still the protocol would work.

In the experimental section, the authors test the setup in a free-space quantum channel with varying levels of loss and compare it to the expected performance in fiber. However, why was the experiment not performed directly using optical fiber? Are there challenges related to coupling into the fiber?

We thank the reviewer for this question. There are no fundamental limitations for QD sources to be coupled to optical fibers, while achieving highest coupling efficiencies is still challenging. But in fact, in our experiment we also do couple to single mode fiber, as the electro-optical-modulator that dynamically prepares the polarization states is a fiber-based device. Depending on the exact definition of where our quantum channel starts, the fiber-pigtail at the EOM output can already be considered part of a "hybrid" quantum channel. Our choice to vary the attenuation in the free-space part was a mere practical one, as this required no special fiber attenuator. To further progress towards practical real-world quantum coin flipping in deployed optical fiber networks, future work needs to address the implementation of QD single-photon sources operating at telecom C-band wavelengths to reduce transmission losses, exploit directly fiber-pigtailed QD devices to improve the systems end-to-end efficiency, and actively compensate for polarization drifts in long fiber-links.

The authors highlight the advantage of single-photon sources in terms of a reduced cheating probability. However, the reported difference is around 0.3%, which seems quite small. Considering the significantly easier implementation using coherent states, does this marginal advantage still justify switching to a single-photon source?

The authors thank the reviewer for raising this interesting question. Our work aimed to explore for the first time a quantum coin flipping protocol using single-photon states, and even succeeded in demonstrating a small, but detectable, single-photon advantage. From a scientific, curiosity driven viewpoint, we are of the opinion that the academic answer is yes, it justifies to switch to a single-photon source. From an economic viewpoint, the answer is probably no, as the technological and financial overhead is currently simply too large. In addition, it is noteworthy, that the QSCF protocol we used in our work is a very general one in which no assumptions are made. This results in relatively high cheating probabilities which cannot be significantly smaller than $\approx 89\%$ (with 90% being realized in our work). Using modified coin flipping protocols with added assumptions, e.g. an adversary with a noisy or bounded storage, the cheating probabilities can be reduced further towards 50%. In this context we also refer to the discussion to the questions of reviewer 1 and Figure 2 where we show that maximizing quantum advantage came at the expense of reducing the effective relative single photon advantage (cf. revised Supplementary Information, Note 3). In addition, we added the following discussion about perspectives for future work on protocol side to the section "Conclusions and Outlook" of the revised manuscript:

While the quantum advantage demonstrated in this work is still relatively moderate, due to the general and assumption-free protocol used, we anticipate that our work stimulates further research in new directions of secure quantum communication and computation. For instance, implementing alternative strong [12] or weak [13] coin-flipping protocols, the achievable quantum and single-photon advantage can be comparatively studied for different protocols. Furthermore, it would be interesting to explore how single-photon sources can enhance the performance of protocols in restricted adversarial settings. Examples are scenarios in which the communicating parties have imperfect memories [14, 15] or restricted measurement capabilities [16, 17], potentially allowing one to achieve cheating probabilities much closer to 50%.

References

- [1] Anna Pappa, André Chailloux, Eleni Diamanti, and Iordanis Kerenidis. Practical quantum coin flipping. *Physical Review A*, 84(5):052305, 2011.
- [2] Yang Zhang, Xing Ding, Yang Li, Likang Zhang, Yong-Peng Guo, Gao-Qiang Wang, Zhen Ning, Mo-Chi Xu, Run-Ze Liu, Jun-Yi Zhao, et al. Experimental single-photon quantum key distribution surpassing the fundamental weak coherent-state rate limit. *Physical Review Letters*, 134(21):210801, 2025.
- [3] Lucas Schweickert, Klaus D Jöns, Katharina D Zeuner, Saimon Filipe Covre da Silva, Huiying Huang, Thomas Lettner, Marcus Reindl, Julien Zichi, Rinaldo Trotta, Armando Rastelli, et al. On-demand generation of background-free single photons from a solid-state source. *Applied Physics Letters*, 112(9), 2018.
- [4] Mujtaba Zahidy, Mikkel T. Mikkelsen, Ronny Müller, Beatrice Da Lio, Martin Krehbiel, Ying Wang, Nikolai Bart, Andreas D. Wieck, Arne Ludwig, Michael Galili, Søren Forchhammer, Peter Lodahl, Leif K. Oxenløwe, Davide Bacco, and Leonardo Midolo. Quantum key distribution using deterministic single-photon sources over a field-installed fibre link. *npj Quantum Inf.*, 10(1):2, January 2024.
- [5] Serkan Ates, Imad Agha, Angelo Gulinatti, Ivan Rech, Matthew T. Rakher, Antonio Badolato, and Kartik Srinivasan. Two-photon interference using background-free quantum frequency conversion of single photons emitted by an inas quantum dot. *Phys. Rev. Lett.*, 109:147405, Oct 2012.
- [6] Beatrice Da Lio, Carlos Faurby, Xiaoyan Zhou, Ming Lai Chan, Ravitej Uppu, Henri Thyrrerstrup, Sven Scholz, Andreas D Wieck, Arne Ludwig, Peter Lodahl, et al. A pure and indistinguishable single-photon source at telecommunication wavelength. *Advanced Quantum Technologies*, 5(5):2200006, 2022.
- [7] Jonas H. Weber, Benjamin Kambs, Jan Kettler, Simon Kern, Julian Maisch, Hüseyin Vural, Michael Jetter, Simone L. Portalupi, Christoph Becher, and Peter Michler. Two-photon interference in the telecom C-band after frequency conversion of photons from remote quantum emitters. *Nature Nanotechnology*, 14(1):23–26, jan 2019.
- [8] Paweł Holewa, Andreas Reiserer, Tobias Heindel, Stefano Sanguinetti, Alexander Huck, and Elizaveta Semenova. Solid-state single-photon sources operating in the telecom wavelength range. *Nanophotonics*, 14(11):1729–1774, 2025.
- [9] Oded Goldreich. *Foundations of Cryptography, Volume 2*. Cambridge university press Cambridge, 2004.

- [10] Anne Broadbent and Christian Schaffner. Quantum cryptography beyond quantum key distribution. *Designs, Codes and Cryptography*, 78:351–382, 2016.
- [11] Jonathan Barrett and Serge Massar. Quantum coin tossing and bit-string generation in the presence of noise. *Physical Review A*, 69(2):022322, 2004.
- [12] Andris Ambainis. A new protocol and lower bounds for quantum coin flipping. *Journal of Computer and System Sciences*, 68(2):398–416, 2004. Special Issue on STOC 2001.
- [13] Dorit Aharonov, André Chailloux, Maor Ganz, Iordanis Kerenidis, and Loïck Magnin. A simpler proof of the existence of quantum weak coin flipping with arbitrarily small bias. *SIAM Journal on Computing*, 45(3):633–679, 2016.
- [14] R. König, S. Wehner, and J. Wullschleger. Unconditional security from noisy quantum storage. *IEEE Trans. Inf. Theor.*, 58(3):1962–1984, March 2012.
- [15] Ivan B. Damgård, Serge Fehr, Louis Salvail, and Christian Schaffner. Cryptography in the bounded-quantum-storage model. *SIAM Journal on Computing*, 37(6):1865–1890, 2008.
- [16] Louis Salvail. Quantum bit commitment from a physical assumption. In *Proceedings of the 18th Annual International Cryptology Conference on Advances in Cryptology*, CRYPTO '98, page 338–353, Berlin, Heidelberg, 1998. Springer-Verlag.
- [17] Ziad Chaoui, Anna Pappa, and Matteo Rosati. Secure quantum bit commitment from separable operations. *arXiv:2501.07351*, 2025.